# META-GLARE: A Computer-Interpretable Guideline System Shell

**Alessio Bottrighi** [1,2,*] and **Paolo Terenziani** [1,2]

1    Computer Science Institute, DiSIT, Università del Piemonte Orientale, 15121 Alessandria, Italy; paolo.terenziani@uniupo.it
2    Laboratorio Integrato di Intelligenza Artificiale e Informatica in Medicina DAIRI, Azienda Ospedaliera SS. Antonio e Biagio e Cesare Arrigo, Alessandria e DiSIT—Università del Piemonte Orientale, 15121 Alessandria, Italy
*    Correspondence: alessio.bottrighi@uniupo.it

**Abstract:** Computer-interpretable Guideline (CIG) systems are important tools for ensuring healthcare practice quality and standardization. They usually provide a tool to acquire CIGs, and one to execute them on specific patients. Current CIG systems rely on their own formalism to represent clinical guidelines, so moving to new phenomena/domains may require substantial extensions. We propose an innovative approach, providing a "shell" that facilitates system designers to define new CIG systems (or to update an existing one) through the definition of a new CIG representation formalism, based on the Task-Network model. We based it on our previous work on META-GLARE, and we extend it with a general execution tool, able to operate on any CIG representation formalism acquired through the META-GLARE acquisition tool. Developed with modularity and compositionality principles, the tool exploits an open library of basic execution methods. It offers a general execution mechanism supporting various CIG formalisms. We successfully applied our approach to three practical case studies. We have identified a reference CIG formalism (the one currently supported by the META-GLARE library) and compared its expressiveness to benchmark approaches. META-GLARE constitutes the first shell in the literature to facilitate the (formalism-based) design and development of CIG systems, considering both acquisition and execution.

**Keywords:** decision support systems; computer-interpretable guidelines; knowledge representation formalisms for clinical guideline; shell to support fast design and prototyping

## 1. Introduction

Clinical practice guidelines (CPGs) are defined as "systematically developed statements to assist practitioner and patient decisions about appropriate healthcare for specific clinical circumstances" [1]. They encode large pieces of medical knowledge and are introduced to optimize the cost of healthcare while improving its quality by putting evidence-based medicine into practice. Starting from the 1980s, the medical scientific community has produced thousands of CPGs. For instance, the Guideline International Network (http://www.g-i-n.net, accessed on 30 May 2023) provides a library consisting of more than 6500 CPGs and includes more than one hundred organizations all over the world. Computer programs to acquire, represent, and apply CPGs in the form of **computer-interpretable** guidelines (CIGs) have been introduced as a tool to facilitate the adoption of CPG in clinical practice, providing several advantages (e.g., automatic connection to the patient databases, and decision-making support). Many different CIG systems have been developed to this end (consider, e.g., [2–5]). The paper [6] provides a comparison of several of such systems (i.e., Asbru, EON, GLIF, GUIDE, PROforma, and PRODIGY) and has also been extended [7] to consider GLARE and GPROVE.

Such works demonstrate that most of the CIG approaches in the literature present several common aspects. Considering the CIG *representation formalism*, the **Task-Network**

**model (TNM)** has been adopted by most approaches. TNM allows one to model the control flow of CIGs as a (hierarchical) **graph** (network) in which nodes represent tasks/actions and arcs represents their flow of control. In turn, nodes and arcs in the TNM are described by a set of properties, or **attributes**. From the *architectural* viewpoint, CIG systems are usually characterized by two main modules: an ***acquisition tool***, supporting the acquisition (in the system formalism) of CIGs for the treatment of a disease, and an ***execution tool***, which takes a CIG (e.g., the CIG for asthma) and a specific patient as inputs and supports the execution of the CIG on the patient. Each CIG system has its own representation formalism, and ***both its acquisition and its execution tools are specifically geared towards the treatment of such a formalism*** (see the surveys [6,8] and the description of the execution engines of different CIG systems in Section 7.1 of this paper).

Such a general approach has a main drawback: whenever the CIG formalism needs to be updated, the whole system must be revised. The software code of both the acquisition and the execution tools must be analyzed to find and revise the pieces of code dealing with the updated part of the formalism. Due to the dimension and complexity of real CIG systems, such update operations may be quite complex and time-consuming (at least, this is our long-term experience with the GLARE--Guideline Acquisition, Representation and Execution—system [9]). In addition, building a new system based on a different formalism is even more time-consuming. In this paper, we overcome such a general drawback by proposing a new methodology and a new architecture to support it (called META-GLARE), to facilitate the design and the fast prototyping of new CIG systems, or the update of existing ones, through the definition of a new CIG representation formalism, based on the Task-Network model.

### 1.1. The Importance of Fast Design and Prototyping of CIG Systems

Notably, especially in the research field, an approach supporting fast design and prototyping when changing a CIG formalism, or proposing a new CIG formalism, can provide crucial advantages, at least in three different types of situations.

First, when approaching new domains or phenomena, the formalism (and consequently the acquisition and execution tools) may need to be extended. Indeed, the history of many CIG systems in the literature shows the need to evolve from the initial version, extend the initial formalism, and/or overcome some of its limitations. For instance, the first version of GLIF (GuideLine Interchange Format) was proposed by a consortium involving Columbia University, Harvard University, and Stanford University in 1998 [10]. A second version, called GLIF2, appeared in 1999 [11], enriching GLIF with an execution engine. To overcome some of the limitations of the expressiveness of GLIF2 formalism, including the lack of a specification for the logical expressions, and the limited set of decision models, a third version (GLIF3) was proposed in 2004 [12]. Similarly, Asbru has been continuously developed, providing different versions (the last one we are aware of is version 7.3) [13]. The same has happened with GLARE. In the original version [14], GLARE supported only "score-based" decisions, while "Boolean" decisions were only introduced at a later stage. More recently, when facing guidelines about *alcohol-related disorders*, we have needed to tackle the issue that different healthcare agents need to be involved, so that the description of each action had to be enriched with different attributes, to model, e.g., the role and qualification required to be the agent in charge of such an action [15]. Recently [16], we have proposed further constructs, such as the possibility of representing complex action synchronizations (e.g., n actions will start when m of their predecessors' end) taken from workflow patterns [17].

Second, to improve the acceptance and the use of CIG systems (and, more generally, computer decision support systems) by physicians, goals like formalism *suitability* and *personalization* are of primary importance. When acquiring a new CIG, *suitability*, intended as the "*intuitive notion of expressiveness which takes the modelling effort into account*" [18], is of primary importance. Notably, there is a *trade-off* between the expressiveness of a formalism and its suitability: if the language is not expressive enough, the domain experts cannot

model the desired phenomena (or have to devolve a lot of efforts to find "tricky ways" to model them), but if it is too rich, they have to consider and select among many different (and mostly not relevant for modeling the specific phenomena) constructs, thus devolving a lot of unnecessary efforts [19].

A very related issue is (formalism) *personalization*: physicians really appreciate the possibility of "representing phenomena as they want", so the possibility of easily prototyping new systems based on "personalized" formalism provides crucial advantages in the CIG context.

Last but not least, although the primary goal of most CIG systems (including GLARE) is *decision support* for physicians, CIG can also be considered for other tasks (e.g., *education*—see Experiment 3 in Section 6; a posteriori cost/quality *evaluation* of patient treatment performed at the organization level, etc.), requiring the representation of different aspects (and for which part of the usual description of CIGs may be irrelevant). In addition, in such cases, the possibility of easily moving to a different formalism can provide crucial advantages.

*1.2. An Approach Supporting Fast Design and Prototyping*

New CIG methodologies to support the design/update of CIG systems, achieving fast prototyping, have been already devised in the area of Artificial Intelligence in Medicine. A major step forward toward such a goal has been proposed by the Protégé [20] and DeGeL [21] approaches. Despite relevant differences, both Protégé and DeGeL are characterized by the fact that they support the acquisition and management of *more than one ontology*. Such ontologies may have different purposes. Specifically, they can be used to model CIG formalisms. In such a way, Protégé and DeGeL support the introduction of a new CIG formalism, or the modification of an existing one, by providing a tool to acquire a new ontology (in our context, a new CIG formalism). Notably, whenever a new ontology (CIG formalism) is acquired, Protégé and DeGeL automatically provide the tool to acquire specific CIG instances formalized in the given ontology (CIG formalism).

As a consequence, if a system designer wants to use Protégé or DeGeL to build a new CIG system (based on a new formalism), or to change the formalism of an existing system, s/he can take advantage of the tool to acquire a new ontology to define the new CIG formalism, and will have the tool to acquire the CIGs described using the new formalism "for free" (i.e., with no programming effort). As a consequence, changing the CIG formalism is facilitated. However, while both Protégé and DeGeL support the acquisition phase, they do not support execution, in the sense that when moving to a new CIG formalism, the system designers will still have to *define/revise the execution tool*. Producing a new execution tool (or modifying an existing one) in response to a change in the CIG formalism is, in general, complex and time-consuming.

To overcome such a limitation, we move the state of the art a further step forward by proposing META-GLARE, a wholly innovative approach to CIGs based on the notion of meta-programming. META-GLARE can be seen as a substantial generalization of the support provided by Protégé and DeGeL to change the CIG formalism, since it also considers the issues related to the execution tools. META-GLARE is suited to cover CIG formalisms based on the general TNM model and following the general assumptions discussed in Section 2. Basically, META-GLARE can be seen as the first "shell" to facilitate the design/revision of GIG systems, considering *both* acquisition and execution.

First of all, META-GLARE provides a tool to acquire such CIG formalisms, and to explicitly represent and store them. Given any acquired formalism F, META-GLARE can automatically produce the corresponding tool to acquire the CIGs expressed in F. However, in META-GLARE, the tool to execute CIGs (expressed in F) is also provided "for free". As we will see, in summary, such a result has been achieved by making both the acquisition and the execution tools of META-GLARE ***parametric with respect to the TNM-Based CIG formalism***. Thus, there is no need to modify them when the CIG formalism is changed, except in exceptional cases (see the discussion in Section 9). As a consequence, META-GLARE achieves fast design and prototyping, both when extending/modifying a

system (i.e., the system formalism) and when designing and developing a new system (i.e., system formalism).

### 1.3. A Glimpse into the META-GLARE Methodology

The methodology we propose to achieve such a goal is distinguished between two main parts of the system:

(1) A part to acquire CIG formalisms. It consists of a module that is similar to the ontology acquisition tools in Protégé and DeGeL, and supports users/system designers in the acquisition of a new formalism, and in its storage (in XML format) in a dedicated and structured library.

(2) A part to *interpret* CIGs. Such a part consists of two main modules: (2.1) a CIG-acquisition module, which supports users in the acquisition of a specific CIG (e.g., a CIG for the treatment of asthma) based on the formalism when it is given a formalism (which is an ***input*** for the module); (2.2) a CIG-execution module which supports the execution of the CIG on the patient when it is given a formalism as input, a specific CIG in the formalism (e.g., the CIG for asthma), and the data of a patient (e.g., John).

In our previous publications, we have already described META-GLARE modules (1) and (2.1) [16,22]. On the other hand, the definition of a module like 2.2 is totally new in the CIG context and is the focus of this paper. The basic feature of the CIG execution module, which makes it unique in the CIG literature, is that it is ***parametric*** *with respect to the CIG formalism*. It may operate on any CIG formalism which can be acquired through the META-GLARE CIG formalism acquisition module. The methodology we propose to devise such a parametric executor is to enforce strict ***modularity*** and ***compositionality*** in our approach. The CIG executor exploits the commonalities between all the CIG formalisms managed by META-GLARE, i.e., the fact that in all such formalisms, a CIG is a *graph* of *nodes* (actions) and *arcs* and that each node and arc is described in terms of a set of *attributes*. Thus, the core idea is that the execution engine navigates the graph (nodes and arcs), executing the *control* attributes for each traversed entity (node or arc) composing it. Importantly, as we will see later, not all attributes of the nodes/arcs have an impact on the execution of a CIG (e.g., the attribute "name" contains just a text string and does not affect execution. We term the attributes that affect execution "control" attributes. In turn, each attribute is typed and, for each type of attribute, the library (built at the formalism acquisition time) contains the methods to execute it. Such methods are invoked by the executor. Additionally, we also propose a *library* of basic types of attributes, which can be exploited when defining a new CIG system through META-GLARE.

Thanks to such an innovative methodology, the *META-GLARE execution module takes CIG formalisms as input*, which has the major impact that, in case the input formalism changes, it still works on it without requiring any modification. In such a way, META-GLARE grants fast design and prototyping whenever the treatment of a new domain requires an extension of the given CIG formalism or the definition of a new one.

### 1.4. Organization and Main Contributions of the Paper

In this paper, we focus on META-GLARE, the first shell in the literature to facilitate the formalism-based design and development of CIG systems, considering both acquisition and execution. Specifically, in this paper, we focus on the execution engine which we have not detailed in any previous publication. However, to make the paper understandable and self-complete, we propose a background section (Section 2), in which we briefly overview our previous work on this topic (see [16] for more details). Sections 3–5 contain the main original and new "technical" contributions of the paper:

- ***Library of control attributes/methods***: a rich library of *control attribute types* (i.e., the basic components of the META-GLARE execution engine; Section 3). A main contribution of our paper is the detailed description of a library of basic methods (associated with constructs of CIG formalisms) invoked by the general execution engine. Such a contribution is proposed in Section 3.

- *Execution engine*: the algorithms constituting the basis of the META-GLARE execution engine (Section 4). The preliminary execution algorithm described in Section 3 of [16] (briefly described in Section 2.3 of our paper) is not very detailed and, more importantly, is very different from the detailed algorithms we now propose in Section 4 (e.g., it is based on the adoption of an execution tree, and such a data structure is not used any more in the algorithms presented in this paper).
- *Evaluation of expressiveness:* an evaluation of the expressiveness of our current formalism (i.e., the more extended formalism supported by our current library) using the benchmark used in [23] to compare four outstanding research approaches in the CIG literature (Section 5). This is one of the most significant contributions of our paper and is entirely new.

Section 6 contains some experiments we ran to evaluate META-GLARE. Section 7 contains related works and comparisons, Section 8 describes our future works, and Section 9 presents conclusions.

## 2. Background

To make the paper self-contained, in this paper, we briefly overview our previous work about META-GLARE. In Section 2.1, we sketch META-GLARE's overall architecture. In Section 2.2, we describe META-GLARE's "meta-formalism". Finally, in Section 2.3, we briefly discuss the *preliminary* previous work that we have devoted to the definition of the execution tool in [16,24].

### 2.1. META-GLARE Architecture

META-GLARE is based on the general TNM model. Specifically, it supports any CIG representation formalism based on the main features (1) and (2) below:

(1)　CIGs are modeled by **hierarchical** graphs, composed of **nodes** and **arcs**.

　　Notably, in META-GLARE, **any type** of **nodes** and **arcs** can be used, provided that

(2)　each **type** of node and arc is defined as **a list of attributes**.

Additionally, META-GLARE does not impose any constraint on the types of attributes that can be introduced in a specific formalism. For the sake of clarity, however, it differentiates attributes into two main categories: the attributes that affect execution (e.g., decision attributes), termed **control** attributes, and the attributes that do not affect it, termed **non-control** attributes (e.g., textual attributes).

Thus, the META-GLARE *interpreter* (CIG acquisition and execution tools) only assumes that a CIG is a hierarchical graph, and it is *parametrized* on the **types of nodes** and **arcs**, and on the **types of attributes**. Regarding execution, META-GLARE assumes that the execution of a CIG can be decomposed into the execution of its nodes (in the order defined by the control arcs) and arcs. In turn, each node/arc is executed by interpreting, in sequence, all the attributes describing it. To support such a general interpretation mechanism, *each attribute type must specify the methods used to acquire and visualize it* and, in the case of control attributes, also *the methods used to execute it*. Such methods are then automatically applied when interpreting a specific CIG. For example, when acquiring a CIG in a given CIG formalism, the META-GLARE acquisition tool acquires the hierarchical graph representing it and applies the methods specified in each attribute type definition in order to acquire the attributes defining each node and arc in the graph. Notably, visualization, acquisition, and execution methods in META-GLARE can be parametrized along two different dimensions: tasks and user type. Such a feature supports task/user-dependent methods (e.g., patients may have different visualization of CIGs with respect to physicians). In the following, we assume the "standard" task (i.e., decision support) and the "standard" users (i.e., physicians). The "testing-for-education" task will be briefly considered in Experiment 3, Section 6. Figure 1 (adapted from [16]) represents a simplified version of the META-GLARE architecture (for a more extensive description, see [16]), with a specific focus only on the execution components. The definition of a new CIG formalism can be provided by system

designers through the DEFINITION_EDITOR, which supports them in the definition of its (i) attribute types, (ii) node/arc types, and possibly (iii) graph constraints. The output of the DEFINITION_EDITOR is an internal XML representation, which is stored in the system's libraries. The HG_INTERPRETER (HG stands for "hierarchical graph") consists of two components, the HG_ACQUISITION module and the HG_EXECUTION module, and manages the generalities of TNM models. In particular, META-GLARE can deal with all the features which are common to all the formalisms. META-GLARE has been implemented by Java Applets. As a consequence, META-GLARE is a cross-platform application. It can be embedded into a web page and can be executed through web browsers without any installation phase. Libraries (see Figure 1) are implemented in PostgreSQL.

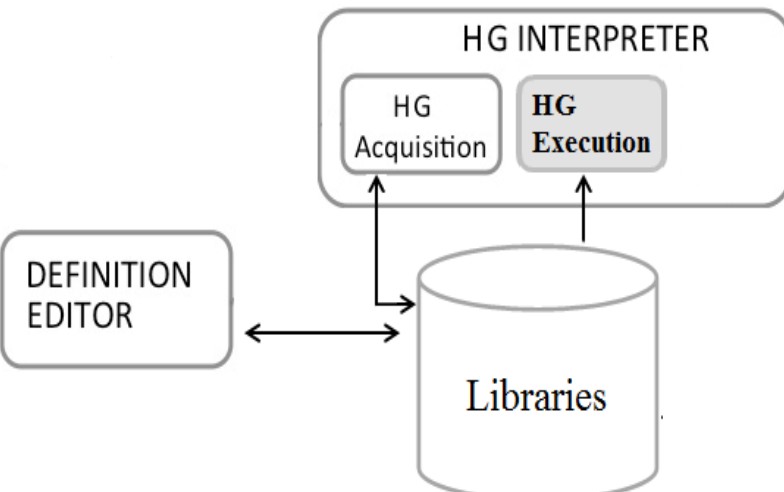

**Figure 1.** The architecture of META-GLARE. HG stands for "Hierarchical Graph" (adapted from [16]).

In this paper, we focus on the META-GLARE *execution* engine, i.e., on the description of the *control attribute types* (i.e., the types of attributes that affect execution; see Section 3) and on the *HG_EXECUTION* module (see Section 4).

Notably, as graphically shown in Figure 2, META-GLARE supports three phases in the CIG system life cycle:

(i)     the acquisition of a CIG formalism (see Figure 2A). In such a phase, the system designer, possibly together with some domain experts (in case they want to "personalize" the CIG formalism; see the discussion in the introduction), can take advantage of the DEFINITION_EDITOR module to define a new formalism (e.g., the GLARE formalism—see Experiment 1 in Section 6).

Once a specific formalism has been acquired, META-GLARE provides the support to operate as in the case of "standard" CIG systems, supporting phases (ii) and (iii).

(ii)    the acquisition of a specific CIG, using the specific CIG formalism (see Figure 2B). In such a phase, knowledge engineers, in cooperation with domain experts, can take advantage of the HG_ACQUISITION module to acquire a specific CIG (e.g., to acquire the CIG about an ischemic stroke using GLARE formalism—see again Experiment 1).

(iii)   the execution of a specific CIG on a specific patient (see Figure 2C). In such a phase, a physician (or a team of physicians) can take advantage of the HG_EXECUTION module to execute a specific CIG (acquired in a specific formalism) on a specific patient (e.g., to execute the ischemic stroke CIG on a specific patient—see Experiment 1).

Several experiments have been drawn to test the three phases. Three of them have been reported in Section 6.

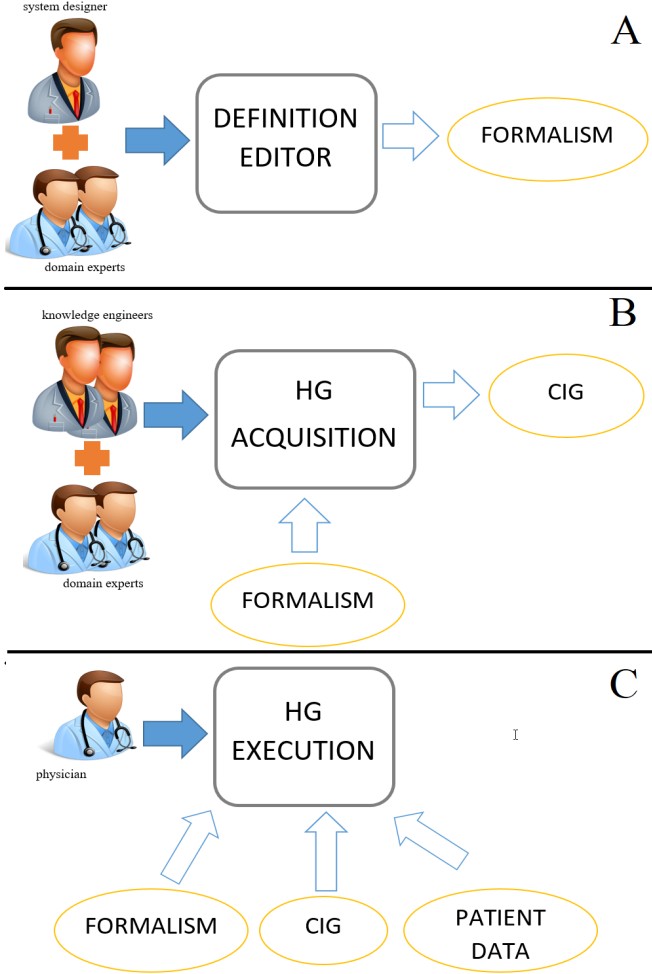

**Figure 2.** The META-GLARE support to CIG system life cycle: (**A**) formalism acquisition, (**B**) CIG acquisition, and (**C**) CIG execution.

### 2.2. A "Meta-Formalism" for CIGs

Using META-GLARE, system-designers can design their own CIG formalism starting from a very general "meta"-formalism (since it is the formalism that we use to define the supported CIG formalisms). In the following, we briefly sketch it (see [16] for more details).

In our approach, we consider TNM-based CIG formalisms (see Section 1): CIGs are represented by hierarchical graphs consisting of nodes and arcs. Each possible CIG representation formalism is characterized by the definition of its types of nodes and arcs (see [16]). In turn, nodes and arcs are characterized by the attributes describing them. In the rest of the paper, we focus only on *control* attributes as non-control attributes do not affect execution and thus are out of the scope of this paper.

#### 2.2.1. Node Types

Each node type represents a general class of actions/tasks (e.g., therapeutic decision), which is characterized by an ordered list of typed attributes.

In particular, the description of node types must include the following attributes:

- the *name* attribute, to specify the name of the node type;
- the *visualization* attribute, to specify the icon graphically representing it;
- the specification of whether the node is *atomic* (not decomposable) or *composed*.

Besides such compulsory attributes, an ordered list of typed attributes can be added. Notice that, in a node, more than one control attribute can be specified.

### 2.2.2. Arc Types

Arc types are defined similarly to node types. In our approach, we have two kinds of arc types: control arc types and non-control arc types. Only control arcs affect the CIG execution flow and will be considered in the following.

In META-GLARE, each arc type is described by an ordered list of compulsory or optional attributes. In particular, an arc type has:

- a *name* attribute, to specify the name of the arc type;
- the *visualization* attribute, to specify how it must be represented;
- a Boolean attribute, which is true if the arc is *oriented*, false otherwise;
- an *ariety* attribute, which specifies the number of nodes connected by such an arc; the ariety is a pair (n1:n2) specifying the number of input (n1) and output (n2) nodes for the arc type. The special value "N" can be used to represent a variable number of nodes n (N stands for a value n $\geq$ 1).
- a typing attribute which specifies the types of the nodes that may be connected by such an arc type. The specifications depend on whether the arc is oriented or not, and on the ariety of the arc.
- A control attribute, consisting of a method that specifies how the arc behaves at execution time.

Other optional attributes are possible (e.g., cost, textual description). In particular, a specific type of optional attribute plays a fundamental role during execution: the attribute temporal constraint. Such an attribute contains the *temporal constraints* between the starting and ending nodes of the arc. More specifically, *bound on differences* constraints [25] between pairs of *endpoints* (i.e., starting and/or ending times of nodes) are modelled in META-GLARE, thus supporting a wide range of both *qualitative* and *quantitative* temporal constraints. The description of our formalism to deal with temporal constraints is out of the scope of this paper and can be found in [26].

### 2.2.3. Attribute Types

In META-GLARE, attribute types are defined using the DEFINITION_EDITOR according to the XML document and are stored as XML files in the library (see Figure 1). Therefore, attribute types can be re-used in different contexts. For example, the attribute type *booleanCondition* can be used to define both pre-conditions and post-conditions of a node type, or in two different node-type definitions (e.g., to define different types of decision nodes). In addition, it can obviously be re-used in different formalisms.

In META-GLARE, each attribute type is characterized by a set of compulsory *features* defining its name, its properties, and its interpretation. Examples of properties are the *mode* (*control* vs. *non-control*) and the syntax which describes the BNF of the possible values of the attribute. The feature "*OntologyType*" is used to support *semantic interoperability*, giving the possibility of linking attribute values to ontological objects. Notably, in META-GLARE, different ontologies may be used in the same formalism (e.g., SNOMED CT [27] for findings and actions, ATC for drugs [28]).

The feature *interpretation* is very important: it contains the pointers to the methods to be adopted by the HG_INTERPRETER in order to acquire, store, and execute any instance of such an attribute type.

In the following, we focus on a few attribute types which are particularly relevant to execution, i.e., the attributes used to define *composite* nodes (i.e., nesting of actions in CIG) and the ones used to model *cycles* of actions.

The attribute type is composed of a control attribute and allows one to specify and store the sub-graph which describes a composite node. This sub-graph is acquired graphically via the HG_ACQUISTION module. In particular, sub-graphs are stored as sets of nodes and arcs in the feature *body*.

In META-GLARE, we propose three different types of control attributes to model cycles: *cycleBooleanCondition*, *cycleNumberRepetition (n)*, and *dynamicCycleNumberRepetition*. In *cycleBooleanCondition* a Boolean condition is used. In the *cycleNumberRepetition (n)*, the

parameter n specifies the number of repetitions (e.g., six repetitions). In *dynamicCycleNumberRepetition*, the number of repetitions has to be dynamically evaluated during the CIG's execution on the basis of a set of patient's parameters. The feature *body* of cyclic control attributes contains a list of control attributes (i.e., the list of operations to be cyclically executed). Bodies can assume two different forms: they may be *composite* actions (control attribute *composed*), or a list of "simple" (i.e., non-composite, and non-cyclic) operations specified through a list of control attributes.

### 2.3. META-GLARE Executor: Previous Work

Although the paper in [16] was only focused on acquisition, for the sake of completeness we have also sketched, at a general level, how the executor could operate. The "essence" of the algorithm for execution in [16] is reported in the following.

Given as input a set of current nodes *currents* in a CIG *G*, the executor considers each current node and executes it by sequentially executing all its control attributes. The execution of an attribute consists of the execution of its execution method. In case such an execution returns an exception, it is managed. When all the control attributes are executed, the control arc exiting the node in the CIG is traversed to get the successor nodes in the CIG. Execution recursively applies to successor nodes. Notably, the "high-level" Algorithm 1 supports the execution of Task-Network models, de-composing the execution of Task-Networks (which are graphs) into the execution of their components (arc and nodes, while nodes, in turn, are constituted by attributes).

---

**Algorithm 1:** Pseudocode of the META-GLARE executor in [16]

---

execute_CIG (set_of_nodes currents, CIG G)
{
**for each** node ∈ currents {
**for each** control_attribute ∈ node {
      out ← exec (attribute.ExecutionMethod)
      **if** (out = exception) **then** manage (out);
    }
Newnodes ← exec (exit_control_arc(node));
execute_CIG (Newnodes, G)
}
}

---

In a subsequent conference paper [24], we have proposed a refinement of the above execution algorithm. However, such a development was from one side quite limited (e.g., it did not support features such as synchronization on n:m arcs—see Section 3.2 below, temporal constraints, etc.) and from the other side unnecessarily complex (e.g., it was based on a data structure, the *execution tree*, which is not necessary). As a consequence, the execution engine presented in Section 4 of this paper is radically different from the one in [24].

In the following, we present the META-GLARE execution engine "bottom-up". In Section 3, we describe the control attributes that we have currently defined in the META-GLARE library. In Section 4, we elaborate on the high-level idea in Algorithm 2 above, detailing the algorithms used to execute the formalism that can be designed in META-GLARE (i.e., the formalism that can be derived from the "meta-formalism" discussed in Section 2 above). Following the principles of compositionality and modularity, we propose three main algorithms, one for graph execution, one for node execution, and one for arc execution.

### 3. META-GLARE Control Attributes

META-GLARE can be conceived as a shell to facilitate the design and development of CIG systems and, in particular, of their execution engine (the focus of this paper). One of the main ingredients of such a shell is a large library of attributes and their control

methods: as shown by the "high-level" informal Algorithm 1, the control methods of attributes are the basic constituents of the execution engines. In order to identify the attributes to be inserted into the META-GLARE Library (for attributes), we have started from GLARE and extended it, considering the many constructs that we have selected after an analysis of several approaches in the literature, with a specific focus on PROforma [29] and Asbru [30]. Additionally, given the similarities between CIG formalisms and workflow formalisms, and the applicability of such formalisms to model clinical guidelines (e.g., [23] and the discussion in Section 7.3), we have also considered selected patterns taken from the Workflow Patterns Initiative [17]. A detailed analysis of the constructs supported by the META-GLARE current library is proposed in Section 5 below.

In META-GLARE, *control attribute types* have a pointer to the method which defines how it must be interpreted by the HG_EXECUTION module (i.e., how their execution will be performed). On the other hand, the HG_EXECUTION module ignores non-control attribute types (for details see Section 5).

In our approach, the execution methods of control attribute types are different depending on whether they can be used to define (i) node types or (ii) arc types. Notably, control attributes provide as **output** a characterization of how the control flow has to be continued. Such a characterization is different, depending on whether the control attribute is related to a node or an arc.

In the following, we describe the main control attributes in the current version of META-GLARE, considering the "standard" task (i.e., execution for decision support). A different task (i.e., execution for testing for education) will be briefly mentioned in Experiment 3 in Section 6.

### 3.1. Control Attributes for Node Types

The control attributes for nodes contain a pointer to the execution method, which contains the Java code to determine the behavior of the node during execution. ***At the current stage***, we have defined the following ***control attribute types*** for ***node types***:

- **composed**: See description in Section 3.3. The execution of a composed attribute type consists of the execution of the sub-graph in its *body* (see for more details Algorithm 3 in Section 4).
- **conditionedAbort**: the evaluation of such an attribute consists in the evaluation of a Boolean condition; if the condition holds, the *abort* modality is returned;
- **conditionedGoTo**: it evaluates a Boolean condition; if it holds, the *goto* modality is returned;
- **conditionedSuspend**: it evaluates a Boolean condition; if it holds, the *suspend* modality is returned;
- **conditionedExit**: it evaluates a Boolean condition; if it holds, the *exit* modality is returned;
- **conditionedFail**: it evaluates a Boolean condition; if it holds, the *fail* modality is returned;
- **unconditionedAbort**: its execution returns automatically the *abort* modality;
- **unconditionedGoTo**: its execution returns automatically the *goto* modality;
- **unconditionedSuspend** its execution returns automatically the *suspend* modality;
- **unconditionedExit**: its execution returns automatically the *exit* modality;
- **unconditionedFail**: its execution returns automatically the *fail* modality;
- **externalInformationAcquisition**: It requires external pieces of information (e.g., patient data from the clinical record). It waits for the availability of the data, or until a message stating that it is not possible to obtain data is received. A failure (*fail* modality) is returned in case the data cannot be obtained;
- **cycleBooleanCondition**: the *body* is repeatedly executed until the Boolean condition holds.
- **cycleNumberRepetition** (n): the *body* is executed n times.

- **dynamicCycleNumberRepetition**: it operates similarly to cycleNumerCondition; the difference is that the number of repetitions is not specified during the CIG acquisition but is evaluated during the CIG execution.
- **suggestion**: it provides some output message to the user (e.g., the textual description of the action or a warning).
- **dataEnquiry**: Looks for one or more data values in the patient database. If values are absent or obsolete, the execution waits until new values are added into the database.

The *output* of the execution of a control attribute for nodes is a *modality*, which may assume the value "**DONE**" (in case the execution succeeds; in such a case, no modification of the control flow must be managed), or values describing how the control flow of the graph has to change. *At the current stage*, we have defined the following modalities:

- **suspend**: the execution (of the CIG) must be suspended until a specific condition becomes true;
- **goto <node>**: the execution must continue by executing <node>;
- **abort**: the execution must be stopped, and terminates;
- **exit to <cig, node>**: the execution of the current CIG must be stopped. The execution restarts from the execution of <node>, which is a node belonging to a different CIG <cig>;
- **fail**: the execution is stopped because of a failure (for instance, a laboratory test could not be executed due to the unavailability of the required instrument) and the HG_EXECUTOR applies the general recovering facility.

Notice that both new control attribute types and new modalities can be defined and added without changing the execution mechanism since the execution meta-engine (see Section 4) is very general. To introduce a new modality, the system designer just has to develop a specific module for it and such a module can be easily integrated into the execution meta-engine.

*3.2. Control Attributes for Arc Types*

The arc control attributes contain a pointer to the Java method, which contains the Java code to evaluate:

(i)     when the arc can be executed,
(ii)    the execution of the arc, and
(iii)   how the CIG flow will continue after the arc execution.

Regarding point (i), an arc can generally be executed when the execution of all its input nodes is ended. However, we also want to admit more complex cases, e.g., an arc that has five input nodes but can be executed when the execution of any two of them is terminated. Thus, META-GLARE implements a waiting mechanism which suspends the execution until the required number of input nodes has terminated.

Concerning point (ii), the execution of an arc may consist of different types of operations (e.g., suggestions to the user physicians).

Regarding point (iii), in the case of an arc with multiple output nodes, its control attribute must specify the procedure to select which one of the output nodes will be executed after the execution of the arc. A typical situation is an arc representing an alternative between different output nodes. In this case, the control attribute may model complex cases in which more than one output node is selected for execution (e.g., "two out of five" output nodes). *At the current stage*, META-GLARE consists of the following control attributes for arc types:

- **sequence (for (1:1) arcs)**. Its execution automatically returns the output node of the arc and the temporal constraints on its time of execution, see the discussion below.
- **booleanDecision** (for (1:N) arcs). A Boolean decision consists of a set of alternative nodes, each one associated with a Boolean condition. It evaluates the Boolean conditions and gives the set of alternatives whose corresponding Boolean condition holds as an output.

- **booleanSuggestion (n)** (for (1:N) arcs). It operates similarly to booleanDecision, the difference is that the output alternatives are only suggested to the user, who is free to follow or ignore the suggestions, choosing n of the output alternatives.
- **scoredDecision (for (1:N) arcs)**. A scored decision is a decision based on scores. It consists of a set of triples <alternative node, Boolean condition, score> plus a threshold for each alternative which has to be compared with the scores achieved by such an alternative. It first evaluates the Boolean conditions. If a Boolean condition holds, the scoredDecision execution method adds *score* to the support value of the *alternative node*. Finally, it compares the score obtained by each alternative with its corresponding threshold and provides, as output, the set of alternatives whose score is greater or equal to the threshold.
- **scoredSuggestion (n) (for (1:N) arcs)**. It operates similarly to scoredDecision, the difference is that the output alternatives are only suggested to the user, who is free to follow or ignore the suggestions, choosing n of the output alternatives (given as output with their temporal constraint).
- **randomDecision (n)**: **(for (1:N) arcs)**. The system randomly chooses n of the output alternatives (given as output with their temporal constraint).
- **Decision (n) (for (1:N) arcs)**. The system asks to the user to choose *n* of the output alternatives (given as output with their temporal constraint) and does not provide any suggestion.
- **qualitativeSuggestion (n) (for (1:N) arcs)**. A qualitativeSuggestion consists of a set of alternative nodes, each one associated with a set of values for different parameters (e.g., effectiveness, cost, time, side effects). It shows to the user the values of such parameters for each one of the alternatives. The user has to choose *n* of such alternatives (which are the output of the arc execution).
- **parallelSplit (for (1:N) arcs)**. Its execution automatically returns all the ending nodes of the arc.
- **Synchronization (n) (for (N:1) arcs)**. The parameter n indicates the number of input nodes whose execution must terminate in order to start the execution of the arc (the number of input nodes must be greater than *n*). Its execution returns the output node.
- **synchronize & Split (n)**,
- **synchronize & booleanDecision (n)**,
- **synchronize & booleanSuggestion (n,m)**,
- **synchronize & scoredDecision (n)**,
- **synchronize & scoredSuggestion (n,m)**,
- **synchronize & qualitativeSuggestion (n,m), (for (N:N) arcs)**.

Such control attributes, as suggested by their names, are just a combination of the above operations.

In all cases, the **output** of the execution of an arc control attribute is a not-empty list of pairs <*outputnode*, *tc*>, where nodes *outputnode* are the output nodes of the arc which have been selected for execution and, for each of such nodes, *tc* is a set of temporal constraints indicating when their execution has to be performed. In particular, for each output node, a minimal and a maximal execution time is specified, following the "agenda technique" [31]. Such times are evaluated on the basis of the current time and the temporal constraints on the arcs of the CIG by a temporal reasoner (described in [26]), which is invoked by the execution engine. Notably, the execution engine checks that the actual execution time of nodes respects the temporal constraints, giving appropriate warning to user-physicians.

### 3.3. Concluding Remark

Before concluding, it is important to stress that we have listed the control attribute types in ***the current version of*** META-GLARE. However, more than emphasizing such a list, it is really important to highlight the fact that the META-GLARE approach is fully *modular* and *compositional*, so that users (system designers) can define and introduce new control attributes in the library whenever needed. Very importantly, only "***local***" programming

is needed: once a new type of attribute is introduced in the library, it can be used in any instance of CIG, and the HG_EXECUTION module will automatically execute it whenever needed. In particular, no modification to the HG_EXECUTION module is needed at all.

## 4. CIG Execution Meta-Engine

The HG_EXECUTION module provides a "general" execution engine that can be instantiated for different CIG formalisms. It has three inputs:

(1)   A CIG formalism $F_i$ (i.e., a set of arc/node types, each one described as an ordered list of types of attributes)

(2)   A specific CIG (say $CIG_j$), expressed in the formalism $F_i$

(3)   A specific patient $P_k$ (we assume that $P_k$'s data are stored in the patient DB)

HG_EXECUTION supports the execution of $CIG_j$ on the patient $P_k$ and constitutes the core of the META-GLARE shell to support the definition of different CIG execution engines. Indeed, it is important to stress that, to the best of our knowledge, all the CIG execution engines in the literature are specifically designed to cope with a specific CIG formalism. As a consequence, they take as input only (2) and (3) above. Notably, META-GLARE's execution mechanism is much more general since any CIG formalism (definable on the basis of META-GLARE's "meta"-formalism) is executable. To enforce such a generality, HG_EXECUTION also takes as input (1) (i.e., a CIG formalism). Notably, HG_EXECUTION only assumes that a CIG is a TNM (i.e., hierarchical graph) while it is *parametrized* on all the other aspects of the CIG formalisms, i.e., on *the types of nodes* and *arcs*, and the *types of attributes* describing them.

META-GLARE (meta-) executor is based on a quite simple basic idea:

(i)   to execute a CIG $CIG_j$ expressed in a formalism $F_i$, it executes the nodes and arcs in $CIG_j$, following the control flow of $CIG_j$;

(ii)   the execution of each node/arc is obtained through the execution of their attributes, in the ordering in which they appear in the description of the node/arc type (notably, each node, arc, and attribute in $CIG_j$ must be an instance of a node *type*, arc *type*, or attribute *type* in $F_i$).

(iii)   Non-control attributes are ignored by HG_EXECUTION. Control attributes are executed by invoking their execution method (i.e., the method associated with the corresponding attribute *type*) and managing their output modality.

The execution is parametrized on the selected CIG formalism and the data of the current patient. For brevity, CIG formalism and patient data are omitted in Algorithms 2–4. Notably, Algorithms 2–4 are original contributions of this paper, being a substantial refinement and improvement of the "high-level" skeleton algorithm sketched in [16] and reported in Section 2.3 of this paper.

In our approach, the CIG execution is basically the execution of a Task-Network model (see Algorithm 2 in the following). The *exec_graph* function is the core of the HG_EXECUTION module. It takes as input the set of the current nodes paired with their temporal constraints (variable *currents* in Algorithm 2; initially the first node of the CIG and with no temporal constraint) and the hierarchical graph (variable G) representing the CIG which must be executed.

The execution terminates when *currents* is empty. Otherwise, the execution is started in parallel on each pair <*n, tc*> ∈ *currents* (line 2). Then, node *n* is executed, invoking the *exec_node* function on the node *n* and the temporal constraints *tc* on its execution time (see Algorithm 3). In the case that the execution of *n* is completed correctly (i.e., *exec_node* returns "DONE"), the executor extracts the exiting control arc *a* starting from *n* (line 5). Note that in our approach, each node has at most one exiting control arc.

Then, the executor waits until the required number of input nodes of the arc *a* i.e., (*get_input_card(get_control_attr (a))*) is terminated (a counter is used to store the number of input nodes that have terminated their execution; the function *get_finished_input* returns such a counter) (lines 7 and 8). Such a part of the algorithm is necessary in order to support

the "synchronized" control attributes for the arcs (see Section 3.2) above. Then, the arc *a* is executed via the *exec_arc* function (line 10), which returns the set *nexts* of next nodes to be executed paired with their temporal constraint. The function *exec_graph* is recursively invoked on *nexts* (and *G*). Notably, in the case that *p* does not have an exit control arc (i.e., *a* is set to NULL in line 6), this branch of the parallel execution ends.

The *exec_node* function (see Algorithm 3) performs the execution of a specific node.

First (line 1), the executor checks whether, at the current time, the temporal constraints *tc* associated with the node are satisfied, sending appropriate warnings to user-physicians (the treatment of temporal constraints is out of the scope of this paper, see [26] for details of the methodology used). Then, the execution of a node consists of the execution of all its control attributes, in the ordering in which they appear in the definition of the node. The non-control attributes are ignored since they do not affect the CIG execution.

First, *attr* is initialized to the first control attribute of the node N using the get_first_control_attr function (line 2). Then, a cycle (lines 3–21) is performed to execute all the control attributes of N using the function get_next_control_attr. If N is a composed node (i.e., the type of attr is composed), its components are executed by calling the exec_graph function on the first node of the subgraph representing the component and the subgraph (extracted through the function get_ components) (lines 4–5).

Otherwise, two different situations must be managed: the node can be cyclic (lines 7–17) or non-cyclic (lines 19–20).

In the first case (i.e., the type of attr is cyclic), the content of attr (i.e., a non-empty set of control attributes) is executed while the condition of attr (retrieved through the function get_condition) holds.

If the content of *attr* is composed (i.e., the node N is a composed cyclic node), its components are executed by calling the exec_graph function on the first node of the subgraph representing the component and the subgraph (extracted through the function get_components) (lines 10–11).

Otherwise, the control attributes which must be repeated will be extracted and executed one at a time (lines 13–17). In line 14, the execution method of the control attribute a is executed. The output of such an execution is a modality. If it is "DONE", the cycle will continue normally on the next control attribute (line 17), otherwise, the execution modality will be managed properly by calling the specific method defined to deal with such a modality (line 16), e.g., to manage the suspension or the abortion of the current execution.

---

**Algorithm 2:** Pseudocode of a graph execution

---

(1) exec_graph(set of <node, temporal constraints> currents, graph G)
{
(2) **for each** <n,tc> ∈ currents **do**
    {
(3)    out ← exec_node (n,tc)
(4)    **if** (out = DONE) **then**
            {
(5)            a ← exit_control_arc (n)
(6)            **if** (a ≠ NULL) **then**
                    {
(7)                    **while** (get_finished_inputs (a) < get_input_card(get_control_attr(a)))
(8)                            **do** WAIT;
(9)                    nexts ← exec_arc (a,G)
(10)                   exec_grap h(nexts,G)
                    }
            }
    }
}

---

---

**Algorithm 3:** Pseudocode of a node execution

---

modality ← exec_node (node N, temporal_constraint tc)
{
(1)     temporal_check (tc, current_time)
(2)     attr ← get_first_control_attr(N)
(3)     **while** (attr ≠ NULL) **do**
        {
(4)             **if** (attr.type = composed) **then**
(5)             exec_graph (<get_first_node (get_components (attr)), tc>, get_components (attr))
(6)             **else**
(7)                     **if** (attr.type = cycle) **then**
(8)                             **while** (eval_condition (get_condition (attr)) **do**
                                    {
(9)                                     a ← get_first_control_attr (attr)
(10)                                **if** a.type = composed **then**
(11)                                        exec_graph (<get_first_node(get_components(a)), tc>,get_components(a))
(12)                                **else**
(13)                                        **while** (a ≠ NULL) **do**
                                            {
(14)                                        modality←execute (get_exec_method(a))
(15)                                        **if** (modality ≠ DONE) **then**
(16)                                                manage_mod (modality)
(17)                                        a ← get_next_control_attr (attr)
                                            }
                                    }
(18)        **else**
            {
(19)                    modality ← execute (get_exec_method (a))
(20)                    **if** (modality ≠ DONE) **then**
                    manage_mod (modality)
            }
(21)            attr ← get_next_control_attribute (attr, N)
        }
(22) return (DONE)
}

---

---

**Algorithm 4:** Pseudocode of an arc execution

---

(1) set of <node, temporal constraints> ← exec_arc (arc a, graph G)
{
(2)     exec_method←get_exec_method (get_control_attribute (a))
(3)     next_nodes ← execute (exec_method)
(4)     nexts ← temporal_reasoning (next_nodes, get_temp_constraints (a), G)
(5)     return nexts
}

---

The HG_EXECUTION module contains a set of methods, one for each one of the modalities described in Section 3.1; however, our approach is open and modular, so that the addition of new modalities and new methods of coping with them is possible.

On the other hand, if the type of attr is not cyclic (lines 19–20), the execution method of attr is executed (line 19) and the output modality is managed as described above (line 20).

The *exec_arc* function (see Algorithm 4) performs the execution of a specific arc a. The execution of an arc consists of the execution of its unique control attribute. The execution method of the control attribute of a is extracted (line 2). Then, such a method is executed (line 3). The output of such an execution is a set next_nodes of nodes (selected among the output nodes of the arc a). In line 4, temporal reasoning is then performed in order to infer, given the temporal constraints in the arc (get_temp_constraints (a)) and graph G,

the constraints on the execution of the nodes next_nodes (for more details about temporal reasoning, please see [26]). The pairs *<node, temporal_constraint>* representing the temporal constraints on its execution time for each node in next_nodes are reported as outputs of the function.

## 5. Evaluating META-GLARE

Proposing an evaluation of a tool like META-GLARE is a complex matter since it is basically a "shell" with which to design and develop CIG systems. To the best of our knowledge, there are no analogous shells in the literature and, consequently, no benchmarks, making it impossible to propose "direct" comparisons. Roughly speaking, META-GLARE is a shell supporting software designers/developers in the modification or creation of new CIG systems. As such, it is also quite impossible to propose significant quantitative performance evaluations: designing and developing a new system (with or without META-GLARE) depends on the complexity of the system and the ability of the designers/programmers. As a consequence, concerning "usability and performance", we could only consider a few case studies in this paper and discuss the suitability of META-GLARE to cope with them (see Section 6). Indeed, it could be appropriate and interesting to carry out an experimental evaluation of performance in which two homogeneous groups of CIG system designers and developers (one using META-GLARE and the other acting as a control group) have to define a theoretically well-specified new CIG system, in order to perform a quantitative (time required to build the new system) comparative performance evaluation. Such an evaluation would be very costly and time-consuming and is out of the scope of this paper.

Indeed, even the evaluation of CIG systems themselves is quite complex and controversial. They are mostly tools ***supporting*** experts and knowledge engineers in the acquisition of clinical guidelines and ***supporting*** physicians in their execution/application on specific patients. Thus, "standard" computer science evaluations (e.g., verification of correctness, analysis of performance) do not seem to apply to CIG systems, since both the correctness of the operations being performed and the time used to carry them out deeply depend on the users being supported. Indeed, we are not aware of any of such evaluations in the CIG literature).

In their review, Isern and Moreno have proposed a comparative evaluation of different CIG systems, considering eleven parameters [8]. Such an evaluation also considers the GLARE system. Considering the eleven parameters in [8], META-GLARE behaves exactly like GLARE; the evaluation is reported on the following parameters (here we adopt the terminology in [8]):

1. *existence of a repository of guidelines: Yes*
2. *presence/absence of a tool offering an editor to create and visualize the guidelines: Yes*
3. *formal representation language used for the guidelines: Task Network*
4. *basic elements defined in the guideline representation language: Query, Work, Decision*
5. *if the tool is designed to be deployed as a distributed system: No*
6. *presence of complex coordination elements: Yes*
7. *type of execution engine: Rule-Based*
8. *connection with an electronic medical record: Yes*
9. *ability to integrate the execution engine with an existing clinical management system: Yes*
10. *use of any standard terminology or representation language: XML, ICD-9*
11. *inclusion of security tools to preserve data integrity and authenticate the accesses to the medical data: No*

In the research area of CIGs, there is a wide consensus that the main parameter along which CIG systems should be evaluated and compared is the *expressiveness* of the representation formalism. Several individual or comparative evaluations of CIG formalism expressiveness have been proposed in the research literature. For instance, ref. [6] provides a comparative evaluation of Asbru, EON, GLIF, Guide, PROforma, and PRODIGY. Workflow

patterns are the consensus benchmark to evaluate the expressiveness of the formalisms offered by workflow systems (see [17]). Recently, such a benchmark has also been applied to CIG formalism, in the milestone comparative analysis in [23] which assesses the state of the art in the CIG research area. In Section 5.1 below, we follow such a line of research, and we analyze the expressiveness of the current version of META-GLARE. However, it is worth pointing out that:

(1) META-GLARE supports multiple CIG formalisms (in principle, all the formalisms that can be generated from META-GLARE are "meta-formalisms"). In the evaluation below, we consider the formalism that includes all the features in the library, as described in Section 3 (we call it **META-GLARE^Lib**). Since META-GLARE is open, additional features can be easily added to the library and, therefore, to META-GLARE^Lib.

(2) Although *expressiveness* is certainly important, we strongly believe that "suitability" [19] also plays an important role in the user-friendliness of a CIG formalism (so it may be the case that user physicians prefer a more restricted formalism that is more suited to their specific task/domain). The essential advantage of META-GLARE is that it helps system designers and possibly domain experts in the definition of a system supporting a "suitable" and "personalized" CIG formalism (see the discussion in the introductory section).

Finally, before moving to the evaluation of the expressiveness of META-GLARE^Lib, we want to remark once again that META-GLARE is not a CIG system, but a "shell" to define/modify CIG systems based on the definition of a CIG representation formalism. As such, we think that it is important and appropriate to propose a "qualitative" experimental evaluation of META-GLARE, to analyze the effort needed by system designers to modify an existing CIG system, or to build a new one using META-GLARE. Such an experimental evaluation is reported in Section 6.

### 5.1. Evaluating the Expressiveness of META-GLARE^Lib

In the following, we evaluate the expressiveness of META-GLARE^Lib, considering the "standard" (i.e., physician decision support) task. Following [23], we use the workflow patterns as a benchmark. The workflow patterns have been identified by the Workflow Patterns Initiative [17] and are the most common control constructs provided by the modeling formalisms used by workflow systems. Workflow patterns are considered a standard for examining the suitability of a process language offered by workflow systems. A total of 43 workflow patterns have been identified by the initiative. Their description is out of the scope of this paper, but an interested reader can find a comprehensive description in [17] or [32].

The workflow patterns are divided into categories on the basis of their characteristics:

- *basic control-flow patterns*: describe basic aspects of process control: sequencing, parallel splitting, synchronizing, exclusive selection and simple merging;
- *advanced branching* and *synchronization patterns*: describe in-between behaviors, where some of the paths in a set of paths can be selected for execution and then different modes of continuation are possible;
- *structural patterns*: identify whether the modelling formalism imposes constraints on how processes are structured;
- *multiple instances patterns*: refer to situations where more than one instance of a task may be active at the same time in the same case;
- *state-based patterns*: describe scenarios in a process where subsequent execution depends on the state of the process instance;
- *cancellation patterns*: refer to the situation where either a single task or a group of tasks in a model need to be cancelled.
- *new patterns*: a set of new patterns and the revised patterns belonging to the previous categories, covering the concepts such as triggers, path and thread branching and synchronization, and cancellation.

Workflow patterns have recently also become a standard benchmark in the evaluation of CIG formalisms provided by the CIG research community. In this section, we extended the analysis done in [23], where EON, Asbru, PROforma, and GLIF are considered in the evaluation of META-GLARE.

To make out analysis more credible, more general, and stronger, we have chosen to adopt the methodology and criteria proposed by the benchmark approach in [23] "as they are", and these are briefly reported below for the sake of completeness.

**Analysis criteria/parameters**. We consider the 43 workflow patterns proposed in [23] to evaluate formalism expressiveness.

**Rating**. As proposed in [23], we rate each pattern as follows:

- *supported* (Y) if a CIG system satisfies the criteria for the pattern and provides a direct support for it,
- *partial support* (Y/N), if a CIG system does not provide a construct that directly supports the pattern, but it compensates by offering alternative solutions through elaborate workarounds or by extending its programming capabilities.
- *no support* (N) if a system does not satisfy the criteria for direct or indirect support.

**Rating explanation**. As proposed for EON, Asbru, PROforma, and GLIF, and also for META-GLARE, we propose a brief qualitative motivation for our evaluation of each pattern, explaining for each control pattern how it is supported in META-GLARE^Lib (in particular, referring to the META-GLARE control attributes described in Section 3) or the reasons why it is not supported.

Table 1 in the following shows the extended analyses including META-GLARE and the supporting comparative analysis of META-GLARE^Lib formalism with some of the milestone formalisms in the CIG research literature on a commonly accepted benchmark. On the other hand, for the sake of brevity, the explanations for the ratings for META-GLARE^Lib formalism are reported in Appendix A.

**Table 1.** Support for the control–flow patterns.

| | Asbru | EON | GLIF | PROforma | META-GLARE^Lib |
|---|---|---|---|---|---|
| **Basic control-flow** | | | | | |
| 1. Sequence | Y | Y | Y | Y | Y |
| 2. Parallel split | Y | Y | Y | Y | Y |
| 3. Synchronization | Y | Y | Y | Y | Y |
| 4. Exclusive choice | Y | Y | Y | Y | Y |
| 5. Simple merge | Y | Y | Y | Y | Y |
| **Advanced branching and synchronization** | | | | | |
| 6. Multichoice | Y | Y | Y | Y | Y |
| 7. Structured synchronizing merge | Y/N | N | N | Y | Y/N |
| 8. Multimerge | N | N | N | N | Y |
| 9. Structured discriminator | Y | Y | Y | Y | Y |
| **Structural patterns** | | | | | |
| 10. Arbitrary cycles | N | Y | Y | N | Y |
| 11. Implicit termination | Y | Y | Y | Y | Y |
| **Multiple instances patterns** | | | | | |
| 12. MI without synchronization | N | N | N | N | Y/N |
| 13. MI with a priori design-time knowledge | Y/N | Y/N | Y/N | Y/N | Y/N |
| 14. MI with a priori run-time knowledge | N | N | N | N | Y/N |

**Table 1.** *Cont.*

| | Asbru | EON | GLIF | PROforma | META-GLARE^Lib |
|---|---|---|---|---|---|
| 15. MI without a priori run-time knowledge | N | N | N | N | Y/N |
| **State-based patterns** | | | | | |
| 16. Deferred choice | Y | N | Y | Y | Y |
| 17. Interleaved parallel routing | Y | N | N | N | Y |
| 18. Milestone | N | N | N | Y | N |
| **Cancellation patterns** | | | | | |
| 19. Cancel activity | Y | Y | Y | Y | N |
| 20. Cancel case | Y | N | Y/N | Y | N |
| **New patterns** | | | | | |
| 21. Structured loop | Y | Y | Y | Y | Y |
| 22. Recursion | Y | N | N | N | Y |
| 23. Transient trigger | N | N | N | Y | N |
| 24. Persistent trigger | N | N | Y | Y | N |
| 25. Cancel region | N | N | N | N | N |
| 26. Cancel multiple instance activity | Y | N | Y | Y | N |
| 27. Complete multiple instance activity | Y | N | N | Y | N |
| 28. Blocking discriminator | N | N | N | N | Y/N |
| 29. Canceling discriminator | Y | N | N | Y | N |
| 30. Structured N-out-of-M join | Y | N | Y | Y | Y |
| 31. Blocking N-out-of-M join | N | N | N | N | Y/N |
| 32. Canceling N-out-of-M join | N | N | N | Y | N |
| 33. Generalized AND-join | N | N | N | N | Y/N |
| 34. Static N-out-of-M join for MIs | N | N | N | N | N |
| 35. Static N-out-of-M join for MIs with cancellation | N | N | N | N | N |
| 36. Dynamic N-out-of-M join for MIs | N | N | N | N | N |
| 37. Acyclic synchronizing merge | N | N | N | Y | N |
| 38. General synchronizing merge | N | N | N | N | N |
| 39. Critical section | Y | N | Y | N | N |
| 40. Interleaved routing | Y | N | Y | N | Y |
| 41. Thread merge | N | N | N | N | N |
| 42. Thread split | N | N | N | N | N |
| 43. Explicit termination | N | N | N | N | Y |

In general, all workflows admit multiple concurrent executions (e.g., a workflow considering the admission procedure of a hospital may be executed on several different patients concurrently). The term "MI" (multiple instances) is used to indicate that in the table. On the other hand, by their own nature, CIG executions are regarding a single patient (e.g., each execution copes with exactly one patient). The workflow patterns 28, 31, and 33 include conditions about how multiple entities (e.g., patients) must be managed within the same execution. If we ignore such conditions (which are trivially satisfied in META-GLARE^Lib since each execution concerns a single entity/patient), such patterns are covered by META-GLARE^Lib. Nevertheless, we have chosen to classify such conditions as partially supported to indicate the fact that we (as well as all the other approaches to CIG) do not consider the concurrent execution of CIGs on multiple patients. Notably, in the

analysis of [23], Asbru, EON, GLIF, and PROforma have been evaluated as not supportive of the above patterns (with the motivation that they do not support multiple patients).

Patterns 12, 14, and 15 are ranked as partially supported since they can be obtained in META-GLARE[Lib], but with some limitations with respect to their general definition in [17]. In particular, for pattern 12, while META-GLARE[Lib] requires that at least one node representing a MI must be terminated before moving to the execution of the next pattern, in the general definition [17], the execution can also continue when no MI is concluded. For patterns 14 and 15, we must assume the availability at the acquisition time of a maximum bound on the number of MIs (while in [17] there is no bound). Notably, in the analysis of [23], Asbru, EON, GLIF, and PROforma have been evaluated as not supportive of the above patterns (since the authors do not consider the possibility of providing "limited" support to a pattern).

As shown in [23], PROforma offers support to 24 patterns (22 direct supports), Asbru to 22 patterns (20 direct supports), GLIF to 18 (17 direct supports), and EON to 12 (11 direct supports); our analysis shows that META-GLARE[Lib] supports 25 patterns (17 directly; consider, however, the discussion above). On the other hand, META-GLARE[Lib] provides a set of features that are not considered in the 43 control patterns:

- Control attributes to manage the interaction (i.e., input/output) with the user (i.e., *suggestion*, *dataEnquiry*, *externalnformationAcquisition*, *booleanSuggestion (n)*, *qualitativeSuggestion*, *scoredSuggestion (n)*, *synchronize&booleanSuggestion (n,m)*, *synchronize&scored Suggestion (n,m)*, *synchronize&qualitativeSuggestion (n,m)*). Such an issue is not considered in the 43 patterns above (except pattern 16, i.e., the deferred choice).

Temporal constraints between nodes. While some of the 43 control patterns impose specific temporal constraints between nodes (for instance, pattern 17 imposes that node execution does not overlap in time), we support a generalized treatment of temporal constraints in META-GLARE[Lib] (in the sense that any temporal constraint in the language specified in [26] can be imposed between nodes).

Our analysis shows that the main limitations in META-GLARE[Lib]'s expressiveness are due to the absence of triggers and exceptions (i.e., eight patterns—18, 19, 20, 23, 24, 25, 26, 27—could be supported in the case that triggers can be supported in META-GLARE[Lib]). This is a deliberate limitation of our current approach, which will be amended in our next version (see the discussion about future works in the Section 8).

### 5.2. Some Limitations of the Current Version of META-GLARE

Although it is quite powerful, the current approach has several limitations, some of which we want to overcome in our future work. First of all, we stress once again that META-GLARE assumes a Task-Network representation formalism for CIGs. As such, it cannot help to develop CIG systems based on other representation paradigms such as Arden Syntax.

Considering the Task Network area, in the current version, the META-GLARE executor is based on the principles (i)–(iii) discussed at the beginning of Section 4. They are quite general but do not cover all possible forms of ***parallelism***. In the current version, META-GLARE can deal with parallelism, but only inside the execution of a single CIG process (e.g., we can deal with the execution of two concurrent actions in a CIG, but not with the concurrent execution of two CIGs, or a CIG and an Exception Handler). However, in some cases, a ***higher level of parallelism*** needs to be present to cope with *concurrent processes (not internal to a specific CIG)*.

(1) Concurrent execution of a CIG with a ***monitor***. In particular, such a level of parallelism is needed to monitor patients' data, in such a way as to be able to ***trigger*** and manage exceptions. In our previous work, we have already extended GLARE with a monitor triggering the treatment of exceptions (see [33]) and managing the interactions between the CIG execution and the exception handler execution; we plan to extend META-GLARE with a proper adaptation of such a mechanism. Notably, the addition of a monitoring-triggering mechanism will greatly extend META-GLARE[Lib]

expressiveness, to also cover with eight new patterns—18, 19, 20, 23, 24, 25, 26, and 27 in [17]. After such an extension, META-GLARE$^{Lib}$ would cover 33 out of 43 patterns, overtaking the Asbru, EON, GLIF, and PROforma approaches.

(2) *Distributed and concurrent execution of two or more CIGs*. Such an extension is needed in case more than one CIG must be concurrently executed on a given patient, e.g., to cope with comorbid patients (see Section 7.1 below). In our previous work, we have extended GLARE to cope with comorbid patients (see the discussion in Section 7.1). However, proposing a "meta-level" support to the "merged execution" of multiple CIGs is a very challenging task that we aim to consider in our future work.

## 6. Applying META-GLARE

META-GLARE aims at facilitating the design and development of a CIG system to acquire and execute CIGs for a new formalism or the modification of an existing system due to a modification of its formalism. In the following, we show three experiments to demonstrate META-GLARE applicability. The three experiments have an increasing intrinsic complexity, and have been selected in order to show three relevant uses:

- The application to META-GLARE to add a new node type (not requiring attributes not already present in the META-GLARE library) in an already existent CIG system (produced through META-GLARE)
- The application of META-GLARE to build a new CIG system, in case all the required attributes are already part of the META-GLARE library.
- The application of META-GLARE to build a new CIG system, in case new attributes have to be added to the META-GLARE Library.

Notably, Experiment 2 builds on the result of Experiment 1 (here we have decided to present examples ordering them on the basis of their intrinsic complexity), and both Experiments 1 and 2 had been already performed and are also reported in [16]. On the other hand, Experiment 3 was only discussed as future work in [16].

**Experiment 1.** To show that META-GLARE facilitates the update of a CIG formalism, in the second experiment we have extended GLARE (previously designed using META-GLARE, as discussed in experiment 2 below) by adding a new type of node, consisting of a set of non-control attributes plus three control attributes, "precondition" (of type booleanCondition; to specify the action's preconditions), "body" (of type external_action), and "postcondition" (of type booleanCondition; to specify the conditions holding after the execution of the action).

Since all the above attribute types were already present in the META-GLARE library, the extension was trivial. The system designer has just used the graphical interface of the DEFINITION_EDITOR to acquire the definition of the new node type in a few minutes. No other effort was required.

After the acquisition of the new type of node, we modified (using the HG_ACQUISITION module of META-GLARE, see Figure 1) the CIG about the ischemic stroke to include, when appropriate, instances of the new type of node. We could then directly execute the updated ischemic stroke CIG on the data of a patient. The META-GLARE HG_EXECUTION module correctly executed the CIG on the patient without any need for modification.

Obviously, some programming is needed if a new version of a control attribute must be introduced to achieve a new task (see Experiment 3, part 1), or if a formalism must be extended with new features for which new control methods have to be added to the META-GLARE Library (see Experiment 3, part 2).

**Experiment 2.** As a first case study, we have applied META-GLARE in order to produce a new version of our original system, GLARE. A Master's student, Irene Lovotti, performed the acquisition of GLARE formalism into META-GLARE, using META-GLARE DEFINITION_EDITOR (see Figure 1). In Figure 3, we introduce a screenshot showing the DEFINITION_EDITOR in action, while acquiring the "data request" attribute.

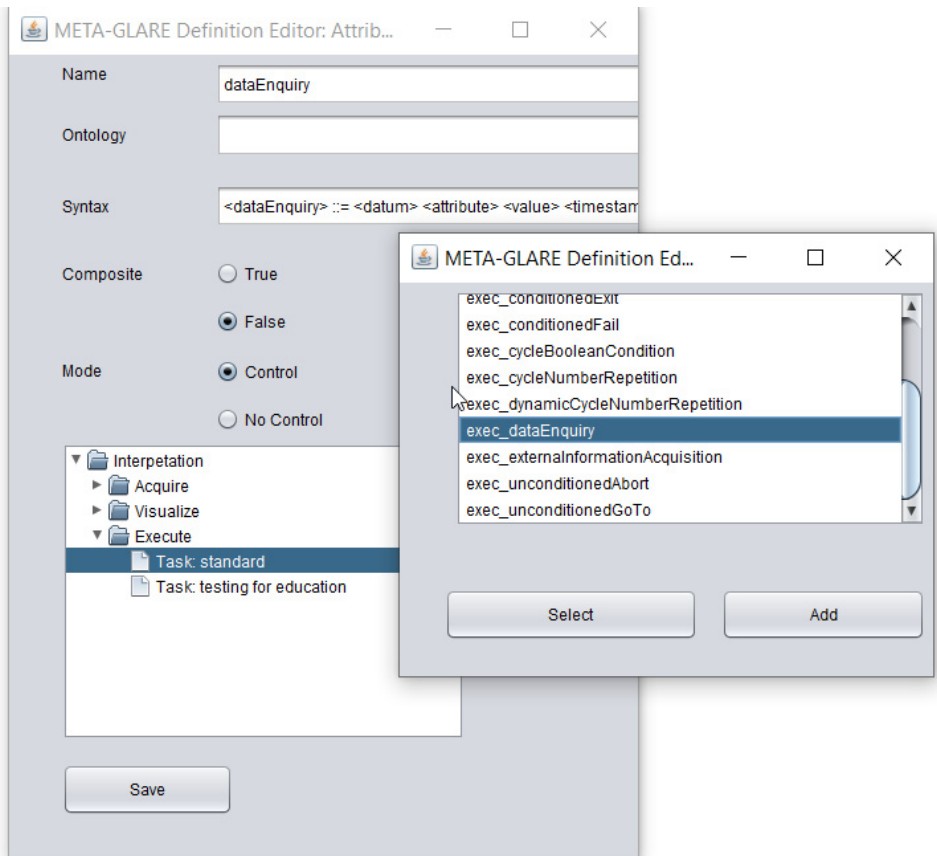

**Figure 3.** A screenshot of the graphical interface of META-GLARE DEFINITION_EDITOR. The screenshot shows a snapshot of the acquisition of the "dataEnquiry" control attribute. Slots (e.g., Name) and pop-up windows are used in order to acquire the different properties of the attribute. In particular, the tree at the bottom of the figure highlights the acquisition of the control method for the "standard" task of "Enquiry". The proper Java method is selected from the library through the pop-up window shown in the right part of the figure.

Such an acquisition required less than one day of work. At the time we performed experiment 1, the META-GLARE library already contained all the attribute types involved in GLARE formalism. Therefore, no additional effort was needed and no programming was needed at all. As a result of the one-day acquisition, we got a new version of GLARE "for free", consisting of both the acquisition and the execution tools.

To continue with the experiment, we considered one of the specific CIGs we had previously acquired with the old version of GLARE (specifically, the CIG about ischemic stroke) and the data of a specific patient and used META-GLARE to execute the CIG on the patient. The META-GLARE HG_EXECUTION module was used and the execution ran correctly with no need for any modification of the HG_EXECUTION module.

In our opinion, this experiment is important to highlight that META-GLARE supports designers in defining a new CIG system (considering both the acquisition and the execution tools) with minimal effort. Indeed, in this specific experiment, no programming at all was needed. However, this was because all the attribute types used in GLARE were already present in the META-GLARE libraries.

**Experiment 3.** META-GLARE for education. "Traditional" courses and books aim at teaching a large body of knowledge to students, e.g., anatomy, disease description and treatment, etc. On the other hand, "operational" knowledge concerning how to operate (diagnose and treat) specific patients is usually neglected in medical texts, and is learned by students only "*by practicing*" [34]. In recent years, several approaches have been proposed to take advantage of computer science to facilitate the teaching of such "operational" aspects,

with a specific emphasis on simulation [34]. However, until now, no approach has tried to exploit the potentialities of CIG systems for education. Indeed, given the data of a real patient or of an invented one, a CIG system like GLARE can be exploited to simulate the execution of any acquired CIG on the patient, to train medical students about how to operate on specific patients. We have started to pursue such a research line in the ROPHS (Report on the Piedmont Health System) project. In such a project, in cooperation with the physicians in the project, we first used GLARE to acquire a guideline for polytrauma. In the second step, the expert physicians have defined a set of "typical" polytrauma patients. Finally, GLARE has been used in a medical course to show students how the polytrauma guideline recommends operating on such patients [33]. The experience has been quite positive and has highlighted the relevance of addressing new educational issues, like the use of a CIG system for testing students.

For testing, given a patient and a CIG, at each step in the diagnosis and treatment of the patient, (i) the student should first specify how she would operate on the patient, (ii) then the CIG system should compare the student's choice to the CIG's recommendation, and (iii) propose the student's discrepancies (if any). In more detail, testing can focus on three main aspects: (1) the student's ability to identify (all and only) the patient's data needed in order to take the decisions regarding the patient, and their ability to make the correct (2) decisions regarding the patient at hand.

To support tasks (i)–(iii), a substantially new (compared to "traditional" systems in the literature) CIG system has to be developed. However, we are taking advantage of META-GLARE to implement it, and this choice is greatly facilitating the achievement of our goal, as sketched below. The implementation of the new system that we have called GLARE-Edu is currently ongoing, following the highlights discussed in [35,36]). Our implementation is organized in two parts.

**Part 1.** In the first (ongoing) phase, we have started from the consideration that, in general, no modification of the chosen CIG formalism is needed when moving from the "standard" task (i.e., support to physicians) to testing for education. As a consequence, we have chosen to maintain GLARE's formalism and to start our implementation of GLARE-Edu on top of the re-implementation of GLARE we have discussed in Experiment 1. Notably, since no change in the formalism has to be performed, no changes to the acquisition methods are needed. Therefore, we have the CIG acquisition tool "for free" from META-GLARE. On the other hand, in the testing-for-education task, the execution of CIGs must be radically different (with respect to "standard" execution). However, thanks to META-GLARE, we do not have to re-implement the whole execution tool. On the other hand, we can exploit the META-GLARE general execution engine as it is, and operate locally, focusing our efforts on the definition and implementation of the methods to support a new *task* (the testing-for-education task) for the control attributes used for the data request (i.e., dataEnquiry, see Section 3.1) and decisions (i.e., booleanDecision, scoredDecision, scoredSuggestion, booleanSuggestion, and qualitativeSuggestion, see Section 3.2).

The execution method for the *testing-for-education task* of the attribute *dataEnquiry* does not directly ask/retrieve patients' findings. On the other hand, it asks the user to list the findings needed at that step of the CIG by selecting them from a pre-defined large list of findings. Such findings are compared with the ones stored in the *dataEnquiry* attribute at the acquisition time to detect unnecessary/irrelevant findings or missing ones.

Analogously, the execution methods for the *testing-for-education task* of decision attributes ask the user to choose between alternative paths on the basis of patients' findings, without providing any hint/form of decision support and evaluate their choice. In all the cases of decisions based on a quantitative evaluation of decision criteria (i.e., *booleanDecision*, *scoredDecision*, *booleanSuggestion*, *scoredSuggestion*, *synchronize and booleanDecision*, *synchronize and booleanSuggestion*, *synchronize and scoredDecision*, *and synchronize and scoredSuggestion*), the decision criteria are then shown to the user, and the user's choice is compared with the decision/suggestion that would be provided by the system by applying such decision criteria to the patients' findings. On the other hand, in the case of *qualitativeSuggestion* and

*synchronize and qualitativeSuggestion*, after the user's choice, the GLARE-Edu simply shows them the qualitative evaluation of the different parameters (e.g., effectiveness, cost, time, side effects) for each one of the alternatives.

**Part 2.** In the second future step, we plan to extend GLARE-Edu to consider also "fake alternatives" [35]. Indeed, for the sake of testing, diagnostic and therapeutic problems might be made more complex for students by adding incorrect alternatives to the clinical guidelines. Such alternatives, not present in the real guidelines, might be significant for testing by representing cases of frequent/plausible medical errors. Evidently, "fake" alternatives must be distinguished from the "correct" ones, and this requires an extension to the CIG formalism (e.g., with the definition of a new 1:n arc in which the exits can be partitioned into "fake" and "correct" ones). However, once again, META-GLARE will support us by allowing us to operate only the new constructs of the formalism locally while still exploiting the general acquisition and execution mechanisms.

## 7. Related Work and Comparisons

In this paper, we have described the META-GLARE execution framework. Therefore, in Section 7.1, we consider some of the CIG execution tools in the literature, with specific focus on Asbru, PROForma, and GLIF3, and on the recent approaches considering the execution of multiple CIGs on comorbid patients. However, it is important to emphasize once again that, while the executors considered in Section 7.1 are specific for a given CIG formalism, the META-GLARE executor is unique since it operates at a higher level of abstraction, supporting the execution of CIGs represented using any formalism that can be generated from META-GLARE's "meta-formalism". In Section 7.2, we show the advantages of our meta-approach with respect to the other approaches in the literature. In Section 7.3, we consider other related approaches in the literature.

### 7.1. CIG Execution Tools

Many CIG systems in the literature provide tools to support the execution of the acquired CIGs. In [8], Isern and Moreno have proposed a comparison between different CIG systems (including GLARE). However, despite the fact that the title of their paper explicitly focuses on CIG execution, very few details are provided about the execution engines of the different systems. Indeed, although most CIG approaches provide an execution engine, very few in-depth descriptions of such engines can be found in the specialized literature. In the rest of this subsection, we focus on three execution engines, which concern three of the most famous CIG approaches in the literature: GLIF, Asbru, and PROforma.

**GLIF** (GuideLine Interchange Format) was proposed in 1998 by a consortium involving Columbia University, Harvard University, and Stanford University to provide a standard guideline representation formalism [10]. Several versions of GLIF have been proposed, finally leading to GLIF3 [12] (see Section 1.3). The GLIF3 Guideline Execution Engine, called *GLEE*, has been described in detail in [37]. GLEE is a client-server system, in which each GLEE client corresponds to the application of a CIG to a particular patient, and a GLEE server supports several tasks, particularly the execution of a specific instance of a guideline on a specific patient. GLEE is based on the "*system suggests, user controls*" philosophy: at any time during the execution of a guideline, users are free to deviate from the guideline recommendations (e.g., not performing the CIG action suggested by the system).

GLIF3 is based on the TNM model. GLIF3 distinguishes between *clinical tasks* (the most important are *action steps*, *decision steps*, and *patient state steps*) and *scheduling tasks* (including *branch step*, *synchronization step*, sequence step, and *subguideline*). The execution of *action steps* depends on the type of task defined in that step. For instance, the execution of a medically oriented action consists of a message to notify the local clinical information system, while a data request task involves data retrieval from the clinical data repository. Two different types of *decision steps* (*case steps* and *choice steps*) are defined in GLIF3. The

execution of a decision *case step* involves the evaluation of the decision criteria associated with each alternative option until one criterion is satisfied. The corresponding option is then automatically selected by the system, and the subsequent step in that option is scheduled for execution. On the other hand, in the case of a decision *choice step*, the alternatives are presented directly to the user (GLEE waits for the user's answer and then schedules the corresponding step). Finally, a *patient state step* models a criterion to define the patient's status and is executed by GLEE's execution engine by asking the user whether the patient meets the criterion or not.

*Scheduling tasks* basically define the *control flow* in the execution of GLIF3 CIGs. In GLEE, the execution of a *branch step* leads to the scheduling and concurrent or any-order execution of subsequent steps. A *synchronization step* in GLIF3 represents a converging point in a CIG's algorithm and is associated with a continuation criterion. The execution of a synchronization step involves the evaluation of such a criterion: the executor waits until the completion of other converging steps eventually leads to its fulfilment. *Sequence steps* simply regulate the order of execution, while the *subguideline* construct supports the hierarchical representation and execution of CIGs.

Notably, the execution of *steps* in GLEE undergoes different execution *states*: (1) the *prepared* state (the execution engine suggests that such a step is executable according to the CIG under execution), (2) the *started* state, (3) the *stopped* state, and (4) the *finished* state.

Finally, GLEE also supports an *event-driven* execution model by defining triggering events for a specific CIG step.

The first version of **Asbru** [30] was proposed in 1998 by a consortium involving the University of Newcastle, Stanford University, and the Vienna University of Technology to provide a time-oriented, intention-based, and sharable language for clinical guidelines. Asbru then evolved into a series of different versions. Asbru is based on the TNM model and represents a guideline as skeletal *plans*. A plan is described by a set of attributes: *preferences*, *intentions*, *conditions*, *effects*, and a *plan body*. *Preferences* define the criteria of applicability. *Intentions* model the goals and are used for critiquing. *Conditions* describe the criteria for which a plan can be started, suspended, reactivated, aborted, or completed. *Effects* define the expected behavior of a plan's execution. The effects can be associated with a probability. The *plan body* for decomposable (i.e., non-atomic) plans contains the set of plans, which consist of the plan and the type of plan, which defines the order of execution of the subplans (i.e., *sequence*, *any order*, *parallel*, *unordered*, or *periodic*). Subplans can be defined asvmandatory or optional for the successful execution of the parent plan. Moreover, Asbru defines two types of non-decomposable plans: *actions* and *user-performed* plans (see below).

*AsbruRTM* (Asbru-Run-Time Module) [38] is the ***execution engine*** framework of AS-BRU. AsbruRTM has three core modules: the data abstraction, environment monitoring, and execution units. Moreover, AsbruRTM provides physicians with a graphical user interface.

AsbruRTM executes a plan by executing all its subplans. In AsbruRTM, the successful execution of parent plans is defined on the basis of which or how many subplans have to be completed successfully. AsbruRTM schedules the subplans on the basis of the parent plan's type and the attributes of the subplans. Concerning non-decomposable plans, *actions* are executed automatically by AsbruRTM while the execution of *user-performed* plans is performed through an interaction with the user, to register if and when the plan is ended and if it was successful. AsbruRTM provides two mechanisms to implement the decision between alternative plans: (i) in the case of decomposable plans, their filter conditions (i.e., conditions about plan applicability) are automatically evaluated to trigger their execution; (ii) in the case of non-decomposable plans, an "if then else" mechanism is adopted: AsbruRTM automatically evaluates the condition and executes the proper branch. Notably, the execution of a plan in Asbru undergoes different states, regarding plan applicability (i.e., *considered*, *possible*, *rejected*, *ready*) and plan execution (i.e., *activated*, *aborted*, *suspended*, *completed*).

**PROforma** was proposed by Imperial Cancer Research Fund UK in 1998 [29]. In PROforma, a guideline is modelled as a set of *tasks* (hierarchically organized into *plans*) and *data items*. The TNM model is adopted, in which nodes represent tasks (and plans) and arcs model the scheduling constraints between them. PROforma distinguishes among three main basic types of tasks (plans are sets of tasks): *actions* (representing external procedures, such as medications or drug administrations), *enquiries* (data requests), and *decisions*.

The PROforma formalism has been the basis of different system implementations, including Arezzo (commercialised by InferMed Ltd.) and Tallis (by Imperial Cancer Research Fund UK). In particular, Tallis consists of a set of Java components, including an Engine which performs the *execution* of PROforma CIGs. We are not aware of any direct description of any specific implementation of an execution engine for PROforma. However, in [39], an accurate definition of the semantics of PROforma has been given, providing an abstract and formal specification of the properties of the operations that any execution engine for PROforma must satisfy. Specifically, **operational semantics** is adopted: the semantics of PROforma are expressed in terms of state transitions of an abstract machine (called an abstract execution engine). In particular, the state of the abstract engine is described through four components: (i) the *Properties Table*, modelling the current value of the properties of all the components of the guideline, (ii) the *Changes Table*, containing new values for such properties, obtained as a result of the operations performed on the guideline, (iii) a logical flag *Exception*, to signal the occurrence of an abnormal event in the guideline execution, and (iv) an *EngineTime*.

In such semantics, a PROforma guideline is modelled through a set of components: Data Items, Candidates, Arguments, Warning Conditions, Parameters, Sources, and Tasks. In particular, during execution, tasks may assume different states (*dormant*, *in_progress*, *discarded*, and *completed*—the possible transitions between states are defined in [39]).

The basic operation to model the execution of a PROforma guideline is *runEngine*, which repeatedly updates the *Properties Table* to reflect the consequences of performing operations and following the scheduling constraints (e.g., *sequence* of *tasks*) in the CIG. For example, after the confirmation of a task, the *runEngine* operation may cause other tasks to enter the in_progress state as a result of their scheduling constraints being satisfied. Technically speaking, *runEngine* achieves the above effect by repeatedly performing a *Burst* operation and then an *enactChanges* operation until the *Changes Table* becomes empty. *Burst* and *enactChanges* are not public operations (i.e., they cannot be directly performed by external systems). The *Burst* operation examines each task through the *reviewTask* operation and determines whether any changes to the guideline are implied by the state of that task. *reviewTask* evaluates the conditions through the *initialiseConditions*, *startConditions*, *discardConditions*, and *CompleteConditions* operations for the state changes of a task *T*, and, if satisfied, updates the *Changes Table* accordingly through the operations of *initialise*, *start*, *discard*, and *complete*, referring to *T*. *enactChanges* updates the state of the execution engine (specifically the *Properties Table*) in accordance with the changes recorded in the *Changes Table* and removes such changes from the *Changes Table*.

Among the most recent approaches to CIG executions, we mention [40–42]. In particular, ref. [40] addresses the problem of extending the executor to support the coordination of multiple agents in the execution of a CIG on a specific patient, also considering phenomena such as task delegation.

Since META-GLARE supports formalisms based on the Task-Network model, its executor has several similarities to the ones of GLIF, Asbru, and PROforma, such as the computation of the control flow of the execution, which is determined by the arcs in the CIG graph. Additionally, META-GLARE's treatment of modality partly encompasses the state transition model for CIG actions used in GLIF, Asbru, and PROforma. However, there is a fundamental difference between META-GLARE's executor and all the other CIG execution engines in the literature: while the executors of the other approaches are specific to a given CIG formalism, META-GLARE's executor is not. It is designed in such a way that it compositionally operates on the general constructs of the Task-Network model (i.e., nodes,

arcs, and attributes), and is *parametric with respect to the specific formalism*. In particular, the *methods* used *to execute* the specific types of attributes in the formalism are *imported from the formalism definition*. As a consequence, the overall META-GLARE approach can be conceived as a shell to support the design of new CIG systems, based on a CIG formalism or a modification of an existing one. The advantages of adopting such a "high-level" shell are discussed below, also taking into consideration Examples 1–3 in Section 6 above.

Finally, it is worth mentioning that, in the last few years, a new type of CIG execution engines has started to appear in the specialized literature, aiming at supporting physicians in the management of *comorbid patients* [43]. Comorbid patients are patients affected by multiple diseases, so multiple CIGs (one for each disease) should be executed on them. However, unfortunately, there may be interactions between the effects of the actions of different CIGs, and such interactions may be dangerous for the patients. Therefore, a "simple" in-parallel execution of multiple CIGs is not a feasible option, and several approaches have been devised to detect and manage possible interactions and to produce a "merged" therapeutic plan for comorbid patients, avoiding dangerous interactions. While the main focus of [44] is the knowledge-based automatic detection and management of interactions [45–52], there is also a focus on the definition of *CIG execution mechanisms* supporting the run-time application of multiple CIGs to a comorbid patient. Ref. [46] focuses on the generation of an "interaction-free" therapeutic plan for comorbid patients through the mitigation of possible interactions via pre-defined rules based on a logical framework. In [45], the authors propose a different implementation of their approach, on top of PDDL, and also supporting patient preferences. In the approach in [46], the run-time executor performs (when interaction can occur) the "merge" of CIGs locally only, and in a dynamic way, in order to take into account the evolution of the patient and of the contexts (e.g., availability of resources). A similar approach has been also proposed in GLARE-SSCPM [48,49], an extension of GLARE in which physicians can interact with the execution mechanism to choose among possible alternative ways of performing "local" merges of the CIGs, also taking advantage of CSP mechanisms to merge CIG constraints [50]. Constraint satisfaction is also used by MuCIGREF [51]. MuCIGREF is a tool for multiple CIG representation and execution which generates personal care plans from CIGs applied to comorbid patients. It supports the concurrent execution of multiple CIGs, managing concurrency or synchronization relations between activities possibly taken from different CIGs, to avoid care conflicts. MuCIGREF supports dynamic constraint satisfaction over CIG models (through the development of a CSP-solving algorithm).

In the current version, META-GLARE does not provide support for the execution of multiple CIGs on comorbid patients. Such an issue will be addressed in our future work.

### 7.2. The Advantages of META-GLARE's "Meta"-Approach

In the following, we highlight the advantages provided by the adoption of META-GLARE with respect to (1) "traditional" CIG systems (e.g., Asbru [30], EON [53], GLIF [12], GPROVE [54], GUIDE [55], PRODIGY [56], PROforma [29]), and (2) Protégé [20] and DeGeL [21]).

First of all, it is important to emphasize commonalities: META-GLARE, Protégé, and DeGeL can be used in different ways to define a new CIG system based on a specific CIG formalism. However, the behavior of such systems, as well as the "traditional" systems, in terms of how to acquire and execute a specific guideline, is basically the same.

On the other hand, META-GLARE is clearly different from the other approaches in case (i) a new CIG system has to be defined on the basis of a new CIG representation formalism, or (ii) an existing system has to be modified in order to update its representation formalism.

(1) **"traditional" CIG approaches**. Traditional CIG systems are based on a specific CIG representation formalism. The software code of the acquisition and execution tools provided by such a system has been specifically developed to manage the chosen representation formalism. Such systems do not provide any facility to change the representation formalism. Therefore, if some change to the formalism is required

(e.g., in order to face new phenomena that had not been considered in the original formalism), the system designer has to operate directly on the software code of the acquisition and the execution tools, inspecting it and identifying where changes have to be performed before appropriately changing/extending such a code. Notably, such an approach may be quite difficult and time-consuming in real CIG systems, whose code dimensions and complexities are quite high, even in the case of limited changes in the formalism.

(2)  **Protégé** and **DeGeL approaches**. Protégé and DeGeL support a mechanism for the acquisition and management of multiple ontologies, which can be exploited to acquire different CIG representation formalisms. For each acquired CIG formalism, they automatically provide a tool to acquire CIG instances expressed in such a formalism. As a consequence, Protégé and DeGeL facilitate designers in the definition of a new CIG system based on a new formalism, or in the update of an existing one to update its formalism: the designer can use their formalism acquisition tool to define the new/updated formalism and gets the tool to acquire CIG instances automatically. However, Protégé and DeGeL do not support the system designer with regard to the execution tool. When changing the CIG formalism or defining a new formalism, the system designer has to operate directly on the software code of the execution tool, inspecting it, identifying where changes have to be performed, and appropriately changing/extending such a code.

(3)  **META-GLARE approach**. As shown by Experiments 1–3 above, META-GLARE further generalizes the support provided by Protégé and DeGeL to also consider the execution tool. The META-GLARE DEFINITION_EDITOR module supports system designers in the definition of a new CIG formalism, or in the update of an already acquired one. Since both META-GLARE's tool to acquire specific CIG instances (HG_ACQUISITION module; see Figure 1) and the tool to execute them (HG_EXECUTION module; see Figure 1) are *parametric* with respect to the input formalism, the definition/update of a CIG formalism does not require any modification to them. No inspection of their code and no modifications are needed at all. Therefore, programming is needed only in case a new component has to be added to the component library (see Experiment 3 above). Notably, our approach is fully modular, since the system designer has only to program the additional component without taking care of the rest of the system. Therefore, META-GLARE supports fast prototyping of the new or extended system.

### 7.3. Other Related Approaches in Computer Science

The idea underlying the META-GLARE approach is both simple and powerful: to develop a "meta-CIG-system", i.e., a shell to define or modify CIG systems on the basis of the CIG representation formalism one wants to provide. Roughly speaking, the input of META-GLARE is a description of a representation formalism for CIGs and the output is a new CIG system to acquire, represent, and execute CIGs described in such a formalism. Notably, similar ideas have already emerged in computer science. For instance, consider the development of the so-called "compilers of compilers", like YACC (Yet Another Compiler of Compilers [57]) in the 1970s. However, while YACC takes grammar denoting a context-free language as the input and produces a compiler for it as the output, META-GLARE operates on CIG formalisms (input) and provides CIG acquisition and execution tools as output. More recently, Model-Driven Software Engineering (MDSE) has emerged as a new methodology for developing software systems in general, and healthcare systems in particular (e.g., the International Workshop on Metamodeling for Healthcare Systems has existed since 2014 (http://mmhs.hib.no/2014/, accessed on 30 April 2022)). Like the Unified Modeling Language (UML), MDSE relates to models for software development. However, while UML concerns documentation, MDSE models are considered equivalent to code since their implementation is (semi)automated. As in MDSE, we use three levels of models (the meta-formalism level, the formalism level, and the CIG-instance level). How-

ever, in our approach, the model is not used to generate new code: the HG_INTERPRETER is already provided by META-GLARE, which automatically "instantiates" it to support the input formalism (which takes the place of the "model" in MDSE).

In practice, CIGs are a way to model and manage clinical processes. As such, they belong to the very large field of computer science approaches to ***process management***, for which many different families of models and methodologies have been and are being devised. For example, different workflow formalisms such as BPMN [58] and BPEL [59] have been devised to deal with processes (e.g., in the business process context). Although different workflow formalisms have been devised in the literature, one of the common focuses of most of them is process orchestration and resource management [60]. The Workflow Management Coalition has not proposed a specific standard for the workflow *engines* but has stated that the workflow enactment service may be considered as a *state transition machine*, where individual process or activity instances change states in response to external events (e.g., completion of an activity) or specific control decisions (e.g., navigation to the next activity step within a process) [61]. For instance, Egon Borger and Bernhard Thalheim [62] propose a modelling of BPMN workflows into the *Abstract State Machine* model and its execution through a special-purpose *ASM simulator*. In the last few years, specific attention in the area has been devoted to considering two important types of processes: *web services* and *human tasks* (i.e., "work which has to be accomplished by people"). BPEL [59] was one of the first workflow approaches aiming at providing a common execution framework and language to manage distributed business processes based on multiple web services. The Web Services Human Task (WS-Human Task) [63] and the BPEL4People [64] proposals have focused on the additional modelling and management of human tasks. Specifically, the WS-Human Task model has been proposed as an extension of Web Services to manage human tasks in a distributed environment, with specific emphasis on coordination and task assignment. BPEL4People has been proposed by the OASIS WS-BPEL Extension for People Technical Committee as an extension of BPEL to enable the definition of human tasks and human interactions as Web Services [65]. A comparative analysis of the WS-Human Task and the BPEL4People has been proposed in [66].

Since both workflow and CIG frameworks are devoted to managing processes, they are closely related. Indeed, in a milestone paper, the workflow patterns identified by the Workflow Patterns Initiative have been adopted in order to evaluate the expressiveness of several CIG formalisms (see [23] and also Section 5). Within the CIG literature, it has been shown that PROForma [29] can manage hospital workflows as well as patient careflows [67], and GUIDE is based upon a workflow-based formalism and engine [53]. Within the workflow literature, Van der Aalst et al. [68] have shown that the Workflow patterns are expressive enough to also cope with CIG patterns, and [69] proposes the adoption of workflows for healthcare tasks. Notably, although we are not aware of any direct implementation of a CIG execution engine as a state transition machine, the operational definition of the semantics of PROforma (see Section 7.1) can certainly be interpreted as a step towards such a direction. Despite many commonalities, however, CIG and Workflow frameworks also present relevant differences. For instance, in [70], a real-world case study (i.e., a comparative analysis of the workflow and the CIG adopted by a real hospital to manage vinorelbine treatment for advanced non-small cell lung cancer) has been used as a starting point to identify and abstract the main differences between workflows and CIGs. The paper shows that workflows and CIGs *differ along five different dimensions: contents*, *focus*, *goals*, *users*, and *editors*. Other works in the literature emphasize the differences between workflows and CIGs, particularly the fact that while workflows mostly focus on *patterns of processes*, CIGs are mostly centered on the modelling and the execution of *knowledge-based decisions* to support evidence-based medical decision-making [71]. Given such differences, in [70], the authors suggest that workflows and CIGs should be represented and executed by *different* frameworks which need to *interact*, in case both of them are used as a support to treat patients in healthcare organizations.

## 8. Future Works

The implementation of the META-GLARE execution engine (HG_EXECUTION module in Figure 1) is almost completed. In particular, the current implementation is more focused on management *control flows* and not on issues concerning the interaction with the users/physicians. After the next implementation works, we plan to start an extensive experimental evaluation of our approach as soon as possible.

Notably, the current approach shows some limitations as discussed in Section 5.2. In our future work, we also aim to further extend it to support new control structures and to overcome these limitations.

We plan to exploit META-GLARE for fast prototyping and the possibility of sharing attributes in our research and to develop CIG systems addressing new CIG tasks. In particular, we plan to exploit the META-GLARE approach in the field of CIG education; see Experiment 3 in Section 6. We plan to build GLARE-Edu by exploiting META-GLARE (see considerations in Section 6). This research will be done in the AI-LEAP project, a project founded by the Fondazione Compagnia di San Paolo and by Fondazione CDP, Bando Intelligenza Artificiale 2. In addition, META-GLARE will allow us to support "personalized" CIGs in our projects with physicians. This means not only making the CIG systems (built using META-GLARE) more appealing, but META-GLARE will allow us to respond more easily and quickly to specific requests and issues which are domain-dependent and task-dependent.

## 9. Conclusions

The overall goal of our work is the definition of META-GLARE, the first shell in the literature to facilitate the formalism-based design and development of CIG systems. While the general architecture of META-GLARE has already been published in [16], in this paper we focus on its execution engine. From the "technical" point of view, the main original contributions of our paper are:

- The algorithms constituting the basis of the META-GLARE execution engine
- The library of *control attribute types*
- An evaluation of the expressiveness of our current formalism (META-GLARE$^{\text{Lib}}$) using the benchmark in [23]

Additionally, some experiments that we ran to apply META-GLARE to case studies are also presented.

In general, our approach operates in the context of providing information support to the treatment of clinical practice guidelines (CPGs), which are one of the major tools that have been introduced to optimize and standardize healthcare practice, taking advantage of the medical knowledge and evidence-based information about the "best" procedures to cope with diseases. Many Computer-Interpretable Guidelines (CIG) have been developed to support the adoption of CPGs in clinical practice. In this paper, we propose an innovative methodology for the design and development of new CIG systems (or the update of existing ones), and a software framework to support it, META-GLARE, focusing on its execution module (HG_EXECUTION in Figure 1). In our approach, the notion of "meta-programming" has been adopted to achieve a "system to design and develop CIG systems". Indeed, META-GLARE supports *easy* and *fast design* and *prototyping* in the definition of new CIG systems (e.g., Experiment 2 in Section 6) and in the extension of existing systems (e.g., Experiments 1 and 3 in Section 6), as well as the *sharing* of attribute types (and arc and node types) between different CIG systems (for instance, Experiment 3).

Second, although we have demonstrated the "power" (or, in other words, the "expressiveness") of META-GLARE$^{\text{Lib}}$ in Section 5, we aim to overcome the limitations as stated in Section 8. However, it is even more important to stress that the current expressiveness of META-GLARE$^{\text{Lib}}$ is not the true cue point. What is really important is that new control patterns can be easily added to META-GLARE in a modular way. To do so, system developers do not need to modify META-GLARE but just have to define new control attributes (and their execution methods) and add them to META-GLARE's library of attributes. After that,

the new attributes can be easily used in the definition of new nodes and arcs in different CIG formalisms (e.g., Experiments 2 and 3 in Section 6).

**Author Contributions:** Conceptualization, A.B. and P.T.; Methodology, A.B. and P.T.; Writing—original draft, A.B. and P.T. The authors contributed equally to this work. All authors have read and agreed to the published version of the manuscript.

**Funding:** This research had financial support from the Fondazione Compagnia di San Paolo and from the Fondazione CDP, Bando Intelligenza Artificiale 2, AI-LEAP project.

**Acknowledgments:** The authors are also very indebted to Yuval Shahar for highlighting many insights and discussions about our approach and, more generally, about the expressiveness of CIG execution modules.

**Conflicts of Interest:** The authors declare no conflict of interest.

## Appendix A

**Table A1.** Explanation of support for the control–flow patterns in META-GLARE. Control attributes have been detailed in Section 3.

| Pattern | Explanation |
| --- | --- |
| **Basic control-flow** | |
| 1. Sequence | It is supported via an arc type with ariety (1:1) having the control attribute *sequence* |
| 2. Parallel split | It is supported via an arc type with ariety (1:N) and having the control attribute *parallelSplit* |
| 3. Synchronization | It is supported via an arc type with ariety (N:1) and having the control attribute *synchronization (n)*. The parameter $n$ has to be set exactly to the number of input nodes of the arc |
| 4. Exclusive choice | It is supported via an arc type with ariety (1:N) having a control attribute which specifies the decision criteria which has to be evaluated (e.g., *scoredDecision*, *booleanDecision* in Section 3). Notice that, to model the exclusive choice, the decisional must be mutually exclusive. |
| 5. Simple merge | It is modeled by multiple arcs pointing to a single node. |
| **Advanced branching and synchronization** | |
| 6. Multichoice | It is supported via an arc type with ariety (1:N) having a control attribute specifying the choice criteria (e.g., through a *scoredDecision*, or a *booleanDecision*. Notice that the decisional criteria must not mutually exclusive, to support the possible selection of more than one output node. |
| 7. Structured synchronizing merge | META-GLARE does not support this pattern, since META-GLARE control attributes for synchronization require to specify at acquisition time (and not at execution time) the number of nodes to be waited. |
| 8. Multimerge | It is modeled by multiple arcs pointing to a single node. |
| 9. Structured discriminator | It is supported via an arc type with ariety (N:1) and having the control attribute *synchronization (n)*. The parameter $n$ has to be set to 1. |
| **Structural patterns** | |
| 10. Arbitrary cycles | It is supported, since the graph representing a CIG has no structural constraints. |
| 11. Implicit termination | It is supported, since the execution of a CIG in META-GLARE implicitly terminates after that all the enabled nodes have been executed |
| **Multiple instances patterns** | |
| 12. MI without synchronization | It is partially supported, through the adoption of more than one construct. The number $k$ of MI (*multiple instances*) is known at acquisition time. Thus, we can explicitly represent k different instances of action in the CIG. This pattern can be represented through an arc with ariety (1:N) and control attribute *parallelSplit*. Such an arc will have $k$ output nodes, each one representing the same activity. In turn, all such nodes are input nodes for an arc with ariety (N:1) and with control attribute *synchronization (n)*, where $n$ is set to 1. Notably in META-GLARE the execution goes on when the execution of at least one node is ended. |

**Table A1.** *Cont.*

| Pattern | Explanation |
|---|---|
| 13. MI with a priori design-time knowledge | It is partially supported. The number *k* of MI (multiple instances) is known at acquisition time. Thus, we can explicitly represent k different instances of action in the CIG. We can model the pattern with an arc with ariety (1:N) and the control attribute *parallelSplit*. This arc has *k* output nodes, each one representing an instance of the same activity type. In turn, all such nodes are input nodes for an arc with ariety (N:1) and with control attribute *synchronization (n)*, where *n* is set to *k*. |
| 14. MI with a priori run-time knowledge | We partially support this pattern (see discussion in Section 5.1). In META-GLARE, there is no specific construct for multiple instances (of nodes): each instance has to be explicitly represented in the CIG. Thus, the case in which the number of instances is not known at CIG acquisition time (but it is only known at execution time, as in the case of pattern 14) can only be modeled indirectly, through a set of constructs. First of all, a bound *b* on the maximum number of instances must be assumed (this is reasonable, since, anyway, one cannot have an infinite number of instances). Then, a decision (*decision (1)*) between *b* alternatives must be modeled (leading to *b* "dummy" nodes, used only for the purpose of the construction, but not influencing the execution). Each branch *i* of the alternative represents the concurrent execution of $i \leq b$ instances (to model the fact that one can execute one, or two, or . . . . or *b* instances). In each case, the instances can run *concurrently*. This is modeled by introducing a *parallelSplit* arc starting from a dummy node and ending in $i \leq b$ node instances. Then, the different alternative branches must be "merged". This result is achieved in two steps. First, for each branch *i*, we introduce a s*ynchronization(i)* arc (requiring the termination of each one of the *i* instances), ending on a "dummy" node. In turn, a s*ynchronization(1)* arc is introduced, starting from each one of the *b* "dummy" nodes, and requiring the termination of exactly one of the alternative branches (i.e., the selected one). |
| 15. MI without a priori run-time knowledge | It is partially supported, similarly to pattern 14 above. Differently from pattern 14, in Pattern 15 the number of instances is not even known at execution time. We model this case as a random choice between the b alternatives (where b is the maximum bound on the number of instances; *randomDecision(1)* arc). |
| **State-based pattern** | |
| 16. Deferred choice | It is supported via an arc with ariety (1:N) having a control attribute, which specifies the interaction with the user. The following control attributes can be used to this purpose: *scoredSuggestion (n)*, *booleanSuggestion (n)*, *qualitativeSuggestion (n)*, *decision (n)*. |
| 17. Interleaved parallel routing | It is supported using META-GLARE temporal constraints. Using constraints, we can easily specify that concurrent nodes must be executed in any order but without overlapping in time. |
| 18. Milestone | It is not supported, since META-GLARE does not support triggers on the state of the execution of nodes/arcs |
| **Cancellation patterns** | |
| 19. Cancel activity | It is not supported, since META-GLARE does not support triggers. |
| 20. Cancel case | It is not supported, since META-GLARE does not support triggers. |
| **New patterns** | |
| 21. Structured loop | It is supported via a node having one of the following control attributes: *cycleBooleanCondition*, *dynamicCycleNumberRepetition*, *cycleNumberRepetition (n)*. |
| 22. Recursion | It is supported via a node having a control attribute *conditioned/unconditionedGoTo* which points to the node itself |
| 23. Transient trigger | It is not supported, since in META-GLARE does not support triggers |
| 24. Persistent trigger | It is not supported, since in META-GLARE does not support triggers |
| 25. Cancel region | It is not supported. This pattern could be realized using a trigger which can activate a cancel action on a region (a set of nodes). However, META-GLARE does not support triggers. |

**Table A1.** *Cont.*

| Pattern | Explanation |
| --- | --- |
| 26. Cancel multiple instance activity | It is not supported. This pattern could be realized using a trigger that can activate a cancel action, but META-GLARE does not support triggers |
| 27. Complete multiple instance activity | It is not supported. This pattern can be realized using a trigger that can activate a cancel action. However, META-GLARE does not support triggers |
| 28. Blocking discriminator | It is partially supported: the pattern is supported by META-GLARE, but only considering a single patient (see Section 1.3). It can be supported via an arc with ariety (1:N) and with control attribute *parallelSplit*; then all the output nodes of this arc are synchronized via an arc with ariety (1:N) and control attribute *synchronization (n)*, *where n is set to the number of input nodes of the arc.* |
| 29. Canceling discriminator | It is not supported, since in META-GLARE the execution of a branch cannot influence the execution of the other parallel branches |
| 30. Structured N-out-of-M join | It is supported via an arc with ariety(N:1) and with control attribute *synchronization (n)*. |
| 31. Blocking N-out-of-M join | It is partially supported: the pattern is supported similarly to pattern 30, but only considering a single patient (see discussion in Section 5.1). |
| 32. Canceling N-out-of-M join | It is not supported, since in META-GLARE the execution of a branch cannot influence the execution of the other concurrent branches |
| 33. Generalized AND-join | It is partially supported. Indeed, the pattern is supported, but only considering a single patient (see discussion above). An arc with ariety (1:N) and control attribute *parallelSplit* is used to start the concurrent branches; such branches are then synchronized using an arc with ariety (N:1) and with control attribute *synchronization (n)*. |
| 34. Static N-out-of-M join for MIs | It is not supported, since pattern 34 requires that *both* the number of concurrent instances to be executed *and* the number of executed instances to be waited for before the execution can go on is unknown at acquisition time. |
| 35. Static N-out-of-M join for MIs with cancellation | It is not supported, similarly to 34. In addition, instance cancellation is required (and not supported in META-GLARE). |
| 36. Dynamic N-out-of-M join for MIs | It is not supported, similarly to 34. With respect to 34, here the number of instances can randomly vary at execution time. |
| 37. Acyclic synchronizing merge | It is not supported, since in 37 the number of branches to be merged depends on the number of branches previously activated at execution time (only acquisition-time values can be managed in META-GLARE). |
| 38. General synchronizing merge | It is not supported, similarly to 37. |
| 39. Critical section | It is not supported, since META-GLARE does not support the concept of critical section. META-GLARE does not allow to specify conditions on an arbitrary groups of nodes (i.e., nodes not structurally related) |
| 40. Interleaved routing | It is supported via temporal constraints which allow to specify that the nodes belonging to different concurrent branches must be executed without overlapping. |
| 41. Thread merge | It is not supported, since META-GLARE does not support threads. Notice that the concept of thread is not related to the context of CIGs. |
| 42. Thread split | It is not supported, since META-GLARE does not support thread. Notice that the concept of thread is not related to the context of CIGs. |
| 43. Explicit termination | It is supported via the control attribute *unconditionedExit* |

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
