# Peer review of "META-GLARE: A Computer-Interpretable Guideline System Shell"

_applsci, doi:10.3390/app13148164_

Round 1
Reviewer 1 Report
Paper implemented good methodology and presented very important results.
Author Response
Thanks a lot to Reviewer 1 for her/his appreciation!
Reviewer 2 Report
Considering that the section 2 of the article refers to the theoretical foundation (Background), the authors must mention the sources of the figures, whether adapted or not.
Understanding that the authors are the creators of META-GLARE, this reviewer has doubts about how much the algorithms described and referenced are new or original for an effective contribution on the application of META-GLARE, as described in the text of lines 618 to 622.
The authors perform a great comparative evaluation of META-GLARE (Table 1 and section 7.1), but this reviewer has doubts about whether it effectively has a scientific contribution or as a technical evaluation for more commercial publications.
Therefore, I recommend highlighting these points about the text, especially in the general objective of the article, such as example "This article has the main objective of proposing a new application of META-GLARE on oncological diagnoses" or something similar and with a specific focus on the research carried out. .
Author Response
Considering that the section 2 of the article refers to the theoretical foundation (Background), the authors must mention the sources of the figures, whether adapted or not.
====
In the revised version, we have added that Figure 1 has been adapted from Fig.1 in [16]. It is a substantially simplified version of the architecture shown in that paper. Figure 2 is new.
====
Understanding that the authors are the creators of META-GLARE, this reviewer has doubts about how much the algorithms described and referenced are new or original for an effective contribution on the application of META-GLARE, as described in the text of lines 618 to 622.
====
META-GLARE design and development have required quite a long time, and have been carried on in different phases. In [16], we have described the general purpose and architecture of the system, and detailed the meta-formalism and the acquisition tools. As discussed in Section 2.3 of this paper, in [16] we have just sketched the main ideas about the behavior of the execution tool, with the goal of providing readers with a comprehensive idea of how the overall META-GLARE should have worked, when completed. Such a general behavior has been summarized and reported in Algorithm 1 in Section 2 (the “Background” Section) of this paper. Starting from such initial idea, we have designed and developed the current execution tool for META-GLARE, whose description is wholly an original contribution of this paper. Such issues were already discussed in the Background section (Section 2) of the paper. More important, Section 7.4 of the submitted (old) version was fully devoted to highlighting the original contributions of our paper with respect to our previous work. However, to make the originality of our contribution clearer “soon” to readers, we have recalled such issues also when required by this Reviewer, by adding the following sentence (before statements in lines 618-622, old version):
Notably, algorithms Algorithms 2-4 are an original contribution of this paper, being a substantial refinement and improvement of the “high-level” skeleton algorithm sketched in [16] and reported in Section 2.3 of this paper.
======
The authors perform a great comparative evaluation of META-GLARE (Table 1 and section 7.1), but this reviewer has doubts about whether it effectively has a scientific contribution or as a technical evaluation for more commercial publications.
====
Thanks for the appreciation! To address such an important point, in the revised version we have re-formulated the paragraph used to introduce such a comparative evaluation. We have stressed that workflow patterns are a standard benchmark for the Workflow community and, recently also for the CIG research area (indeed, Asbru, EON, GLIF and PROforma, evaluated in [23], are all research products – to the best of our knowledge, only PROforma lead to the production of commercial systems, like Arezzo). The new paragraph is reported also below:
In the research area of CIGs there is a wide consensus that systems should be evaluated and compared on the basis of the expressiveness of the representation formalism. Several individual or comparative evaluations of CIG formalism expressiveness have been proposed in the research literature. For instance, [6] provides a comparative evaluation of Asbru, EON, GLIF, Guide, PROforma, and PRODIGY. Workflow patterns are the consensus benchmark to evaluate the expressiveness of the formalisms offered by workflow systems (see [17]). Recently, such a benchmark has also been applied to CIG formalism, in the milestone comparative analysis in [23], assessing the state of the art in the CIG research area. In subsection 5.1 below we follow such a line of research, and we analyse the expressiveness of the current version of META-GLARE.
Additionally, in response to this comment and to an analogous comment of Reviewer #3, in the revised version we have explicitly discussed the criteria we have used in the comparative evaluation, stressing that we have taken them “as they are” in the benchmark evaluation comparison in [23]. The new discussion is reported below.
Workflow patterns have recently become a standard benchmark also in the evaluation of CIG formalisms provided by the CIG research community. In this section, we extended the analysis done in [23], where EON, Asbru, PROforma and GLIF are considered, to evaluate META-GLARE.
To make more credible, general and stronger our analysis, we have chosen to adopt “as they are” the methodology and criteria proposed by the benchmark approach in [23], briefly reported below for the sake of completeness.
Analysis criteria/parameters. We consider exactly the 43 workflow patterns proposed in [23] to evaluate formalism expressiveness.
Rating. As proposed in [23], we rate each pattern as follows:
- supported (Y) if a CIG system satisfies the criteria for the pattern and provides a direct support for it,
- partial support (Y/N), if a system provides indirect support for the criteria either via extended workarounds or through programmatic extensions
- no support (N), if a system does not satisfy the criteria for direct or indirect support.
Rating explanation. As proposed for EON, Asbru, PROforma and GLIF, also for META-GLARE we propose a brief qualitative motivation for our evaluation of each pattern, explaining (for each control pattern), how it is supported in META-GLARELib (in particular, referring to META-GLARE control attributes described in Section 3), or the reasons why it is not supported.
Table 1 in the following shows the extended analisys including META-GLARE, and supporting a comparative analysis of META-GLARELib formalism with (some of) the milestone formalisms in the CIG research literature, on a commonly-accepted benchmark. On the other hand, for the sake of brevity, the explanations for the ratings for META-GLARELib formalism are reported in the Appendix A.
==========
Therefore, I recommend highlighting these points about the text, especially in the general objective of the article, such as example "This article has the main objective of proposing a new application of META-GLARE on oncological diagnoses" or something similar and with a specific focus on the research carried out.
====
Thanks a lot to this Reviewer for such an important comment. In the revised version, we have deeply revised subsection 1.4 to highlight already at the beginning of the paper the main original contributions. The new version of section 1.4 is also reported below:
1.4. Organization and main contributions of the paper
In this paper, we focus on META-GLARE, the first shell in the literature to facilitate the (formalism-based) design and development of CIG systems, considering both acquisition and execution. Specifically, in this paper, we focus on the execution engine (which we have not detailed in any previous publication). However, to make the paper understandable and self-complete, we propose a background section (Section 2), in which we briefly overview our previous work on this topic (see [16] for more details). Sections 3, 4 and 5 contain the main original and new “technical” contributions of the paper:
- Library of control attributes\methods: a rich library of control attribute types (i.e., the basic components of META-GLARE execution engine; Section 3 ). A main contribution of our paper is the detailed description of a Library of basic methods (associated with constructs of CIG formalisms) invoked by the general execution engine. Such a contribution is proposed in Section 3.
- Execution engine: the algorithms constituting the basis of META-GLARE execution engine (Section 4) The preliminary execution algorithm described in Section 3 of [16] (briefly described in Section 2.3 of our paper) is not very detailed and, more important, is very different from the detailed algorithms we now propose in Section 4 (e.g., it is based on the adoption of an execution_tree, and such a data structure is not used any more in the algorithms presented in this paper).
- Evaluation of expressiveness. An evaluation of the expressiveness of our current formalism (i.e., the more-extended formalism supported by our current library) using the benchmark used in [23] to compare four outstanding research approaches in the CIG literature (Section 5). This is one of the most significant contributions of our paper, and is entirely new.
Section 6 contains some experiments we run to evaluate META-GLARE. Section 7 contains related works and comparisons, Section 8 describes our future works and Section 9 presents conclusions.
Additionally, in the revised version, we have also summarized the main goals and contributions of the paper also at the beginning of the Concluding Section, as reported below:
The overall goal of our work is the definition of META-GLARE, the first shell in the literature to facilitate the (formalism-based) design and development of CIG systems. While the general architecture of META-GLARE has been already published in [16], in this paper we focus on its execution engine. From the “technical” point of view, the main original contributions of our paper are:
- The algorithms constituting the basis of META-GLARE execution engine
- The library of control attribute types
- An evaluation of the expressiveness of our current formalism (META-GLARELib) using the benchmark in [23]
Additionally, some experiments we run to apply META-GLARE to use-cases are also presented.
Reviewer 3 Report
The paper presents an innovative approach to developing and updating Computer Interpretable Guidelines (CIGs) systems. The authors propose a "shell" that facilitates the design and development of new CIG systems or the updating of existing ones. They achieve this by introducing a new CIG representation formalism based on the Task-Network model and extending their previous work on META-GLARE with a general execution tool. The paper emphasizes modularity and compositionality principles and provides an open library of basic execution methods to support various CIG formalisms.
The paper is well-written and structured, providing a clear understanding of the proposed approach. The authors effectively highlight the limitations of current CIG systems, emphasizing the need for a flexible and extensible framework. The integration of the Task-Network model and the open library of execution methods appears to be a promising direction for advancing CIG systems.
One strong aspect of the paper is the successful application of the proposed approach to three practical case studies. This demonstrates the feasibility and effectiveness of the approach in real-world scenarios. However, to further strengthen the paper, more details should be provided regarding the selection and diversity of the case studies. Additionally, including quantitative evaluation metrics to assess the performance of the proposed approach compared to existing benchmark approaches would provide a more comprehensive analysis.
The authors mention that they have identified a reference CIG formalism currently supported by the META-GLARE library and compared its expressiveness to benchmark approaches. It would be helpful to elaborate on the specific criteria used for this comparison and provide a more detailed analysis of the results. This would enhance the credibility of the proposed approach and its advantages over existing methods.
Furthermore, while the paper focuses on the design and development of CIG systems, it would be valuable to discuss any potential limitations or challenges associated with the proposed approach. Addressing these issues would contribute to a more balanced discussion and provide insights for future research in this area.
Minor editing of English language required.
Author Response
The paper presents an innovative approach to developing and updating Computer Interpretable Guidelines (CIGs) systems. The authors propose a "shell" that facilitates the design and development of new CIG systems or the updating of existing ones. They achieve this by introducing a new CIG representation formalism based on the Task-Network model and extending their previous work on META-GLARE with a general execution tool. The paper emphasizes modularity and compositionality principles and provides an open library of basic execution methods to support various CIG formalisms.
The paper is well-written and structured, providing a clear understanding of the proposed approach. The authors effectively highlight the limitations of current CIG systems, emphasizing the need for a flexible and extensible framework. The integration of the Task-Network model and the open library of execution methods appears to be a promising direction for advancing CIG systems.
One strong aspect of the paper is the successful application of the proposed approach to three practical case studies. This demonstrates the feasibility and effectiveness of the approach in real-world scenarios. However, to further strengthen the paper, more details should be provided regarding the selection and diversity of the case studies.
========================
This Reviewer is right: in the previous version, we have not addressed the “significance” of the case studies, and the reasons why we have chosen them. In the revised version, we have emended such a limitation. By the way, by doing this, we also changed the ordering of the presentation, to discuss experiments not in a chronological ordering (as above), but ordering them depending on their “complexity”. In the following, we report the additional discussion we have added in the revised version.
In the following, we show three experiments to demonstrate META-GLARE applicability. The three experiments have an increasing intrinsic complexity, and have been selected in order to show three relevant use-cases:
- The application to META-GLARE to add a new node type (not requiring attributes not already present in META-GLARE Library) in an already existent CIG system (produced through META-GLARE)
- The application of META-GLARE to build a new CIG system, in case all the required attributes are already part of META-GLARE Library.
- The application of META-GLARE to build a new CIG system, in case new attributes have to be added to META-GLARE Library.
Notably, Experiment 2 grounds on the result of Experiment 1 (here we have decided to present examples ordering them on the basis of their intrinsic complexity), and both Experiments 1 and 2 had been already performed, …
=======
Additionally, including quantitative evaluation metrics to assess the performance of the proposed approach compared to existing benchmark approaches would provide a more comprehensive analysis.
While in principle we fully agree with the Reviewer’s comment, and, in particular, with the need to evaluate and assess/compare our approach. However, in practice, we regret to say that none of the “standard” evaluation metrics seems to apply to assess the performance of a “shell”-approach like the one we propose, and we couldn’t get any idea from the literature in our area of research (that we practice since 1996), nor in other areas we know. In our opinion, the core issue is that we provide a shell to support designers/developers of new CIG systems. The performance (e.g., in terms of time required to build a new CIG system) strongly depends on the intrinsic complexity of the (formalism of) the system to be built, and on the abilities of designers/developers. In the revised version, we suggest that, possibly, an appropriate quantitative performance evaluation could consider two homogeneous groups of (CIG) system designers and developers (one using META-GLARE, and the other as control group), provide them the specifications (in terms of formalism) for a new CIG system, and then to perform a quantitative (time required to build the new system) comparative performance evaluation of the two groups. However, such an evaluation would be highly costly (we cannot hire a set of software experts to run the experiment!) and time consuming, and is out of the scope of this paper.
On the other hand, since we fully agree with this Reviewer’s concerns about evaluation, in the revised version we have almost re-written and substantially extended the introductory part of Section 5. For the sake of brevity, here we summarize the main points of our new argumentation.
- Since we propose a shell to support experts to build CIG systems, it is quite difficult to assess quantitatively its performance, since it depends on the ability of the experts, and the complexity of the system to be built. Additionally, since our approach is the first “shell” in the CIG area (including also the execution engine), comparisons with similar approaches are not possible.
- Even if we consider the wide research area of “ground” CIG systems (we propose a shell to produce them), we are not aware of any evaluation of (time) performance [pharenthesis: CIG systems are used to propose a knowledge-based support to physicians in the application of CIGs, while, e.g., the execution time fully depends on the evolution of patients to which the CIG is applied].
- An evaluation of CIG systems has been proposed by Isern and Moreno, considering 11 parameters [8]. In the revised version we have added the evaluation of META-GLARE along such parameters.
- There is a wide consensus in the CIG area that systems must be evaluated and compared considering the expressiveness of their representation formalism. We have considered the benchmark analysis in [23] (which adopts the consensus formalism of workflow patterns [17] as reference formalism to model processes) to evaluate and compare the expressiveness of our approach (i.e., the maximal formalism actually supported by META-GLARE libraries).
In short, in the revised version we now show that our evaluation of META-GLARE is fully in-line with the evaluations proposed in our area of research. We think that assessing such a fundamental issue is a major improvement for our paper, and we thank this Reviewer for her/his inspiring comment.
==========
The authors mention that they have identified a reference CIG formalism currently supported by the META-GLARE library and compared its expressiveness to benchmark approaches. It would be helpful to elaborate on the specific criteria used for this comparison and provide a more detailed analysis of the results. This would enhance the credibility of the proposed approach and its advantages over existing methods.
======
We sincerely thank Reviewer #2 for such an inspired comment. In the revised version, we have clearly stated the methodology we used for the evaluation (Analysis criteria, Rating, Rating explanation), stressing that we have imported them (without any change) from the approach in [23], which fixes a commonly-agreed benchmark for the CIG research community. Additions are also reported below:
Workflow patterns have recently become a standard benchmark also in the evaluation of CIG formalisms provided by the CIG research community. In this section, we extended the analysis done in [23], where EON, Asbru, PROforma and GLIF are considered, to evaluate META-GLARE.
To make more credible, general and stronger our analysis, we have chosen to adopt “as they are” the methodology and criteria proposed by the benchmark approach in [23], briefly reported below for the sake of completeness.
Analysis criteria/parameters. We consider exactly the 43 workflow patterns proposed in [23] to evaluate formalism expressiveness.
Rating. As proposed in [23], we rate each pattern as follows:
- supported (Y) if a CIG system satisfies the criteria for the pattern and provides a direct support for it,
- partial support (Y/N), if a system provides indirect support for the criteria either via extended workarounds or through programmatic extensions
- no support (N), if a system does not satisfy the criteria for direct or indirect support.
Rating explanation. As proposed for EON, Asbru, PROforma and GLIF, also for META-GLARE we propose a brief qualitative motivation for our evaluation of each pattern, explaining (for each control pattern), how it is supported in META-GLARELib (in particular, referring to META-GLARE control attributes described in Section 3), or the reasons why it is not supported.
Table 1 in the following shows the extended analisys including META-GLARE, and supporting a comparative analysis of META-GLARELib formalism with (some of) the milestone formalisms in the CIG research literature, on a commonly-accepted benchmark. On the other hand, for the sake of brevity, the explanations for the ratings for META-GLARELib formalism are reported in the Appendix A.
================
Furthermore, while the paper focuses on the design and development of CIG systems, it would be valuable to discuss any potential limitations or challenges associated with the proposed approach. Addressing these issues would contribute to a more balanced discussion and provide insights for future research in this area
===========00
We fully agree with this Reviewer’s comment. Indeed, some discussion about the limitations of our current approach were already present in the last part of the Concluding section. However, they were “hidden” in the discussion about future work. Therefore, in the revised version, we have revised and extended the discussion about limitations, and we have moved them into a newly created subsection (Subsection 5.2). As a result, Section 5 of the revised version contains a (quite wide) range of evaluations of our approach, including the discussion about its limitations. We thank this Reviewer, since we believe that such a presentation greatly improves the quality of our paper.
Reviewer 4 Report
The paper is well written. The following should be taken care of before publication:
120: ssupportthe typo
781: Table 1. The Y/N notation is not clear; it is explained as partial support, but further explanations should be given
799-821: Provide the text either as notes or as regular text without denoting it as notes
882, 899, 978: Remove the symbol at the end of the sentence
1293: Sections typo
1288-1315: Completely revise section 7.4. New contributions with respect to our previous work to describe what the title denotes and clearly identify the advances of the new system; it currently mostly gives information on past implementations, information that has been already provided earlier in the paper.
1331-1372: Move this part to a separate section ‘Future Work’, before the Conclusions.
1346: Explain what you mean by ‘almost’
Author Response
120: ssupportthe typo
====
Fixed
====
781: Table 1. The Y/N notation is not clear; it is explained as partial support, but further explanations should be given
====
In response to this comment and to an analogous comment of Reviewer #3 and #2, in the revised version we have explicitly discussed the criteria we have used in the comparative evaluation, stressing that we have taken them “as they are” in the benchmark evaluation comparison in [23]. However, we also tried to explain better the Y/N rating. The new discussion is reported below:
Workflow patterns have recently become a standard benchmark also in the evaluation of CIG formalisms provided by the CIG research community. In this section, we extended the analysis done in [23], where EON, Asbru, PROforma and GLIF are considered, to evaluate META-GLARE.
To make more credible, general and stronger our analysis, we have chosen to adopt “as they are” the methodology and criteria proposed by the benchmark approach in [23], briefly reported below for the sake of completeness.
Analysis criteria/parameters. We consider exactly the 43 workflow patterns proposed in [23] to evaluate formalism expressiveness.
Rating. As proposed in [23], we rate each pattern as follows:
- supported (Y) if a CIG system satisfies the criteria for the pattern and provides a direct support for it,
- partial support (Y/N), if a CIG system does not provide a construct that supports directly the pattern, but it compensates by offering alternative solutions through elaborate workarounds or by extending its programming capabilities.
- no support (N), if a system does not satisfy the criteria for direct or indirect support.
Rating explanation. As proposed for EON, Asbru, PROforma and GLIF, also for META-GLARE we propose a brief qualitative motivation for our evaluation of each pattern, explaining (for each control pattern), how it is supported in META-GLARELib (in particular, referring to META-GLARE control attributes described in Section 3), or the reasons why it is not supported.
Table 1 in the following shows the extended analisys including META-GLARE, and supporting a comparative analysis of META-GLARELib formalism with (some of) the milestone formalisms in the CIG research literature, on a commonly-accepted benchmark. On the other hand, for the sake of brevity, the explanations for the ratings for META-GLARELib formalism are reported in the Appendix A.
====
799-821: Provide the text either as notes or as regular text without denoting it as notes
====
Modified according to the request
====
882, 899, 978: Remove the symbol at the end of the sentence
====
Removed
====
1293: Sections typo
====
Fixed
====
1288-1315: Completely revise section 7.4. New contributions with respect to our previous work to describe what the title denotes and clearly identify the advances of the new system; it currently mostly gives information on past implementations, information that has been already provided earlier in the paper.
====
On the basis of such a suggestion and also a similar one of Reviewer #2, we decide to remove section 7.4 and to expand section 1.4 to provide such information to the readers at the begging of the paper.
The new discussion (section 1.4 ) is reported below:
In this paper, we focus on META-GLARE, the first shell in the literature to facilitate the (formalism-based) design and development of CIG systems, considering both acquisition and execution. Specifically, in this paper, we focus on the execution engine (which we have not detailed in any previous publication). However, to make the paper understandable and self-complete, we propose a background section (Section 2), in which we briefly overview our previous work on this topic (see [16] for more details). Sections 3, 4 and 5 contain the main original and new “technical” contributions of the paper:
- Library of control attributes\methods: a rich library of control attribute types (i.e., the basic components of META-GLARE execution engine; Section 3 ). A main contribution of our paper is the detailed description of a Library of basic methods (associated with constructs of CIG formalisms) invoked by the general execution engine. Such a contribution is proposed in Section 3.
- Execution engine: the algorithms constituting the basis of META-GLARE execution engine (Section 4) The preliminary execution algorithm described in Section 3 of [16] (briefly described in Section 2.3 of our paper) is not very detailed and, more important, is very different from the detailed algorithms we now propose in Section 4 (e.g., it is based on the adoption of an execution_tree, and such a data structure is not used any more in the algorithms presented in this paper).
- Evaluation of expressiveness. An evaluation of the expressiveness of our current formalism (i.e., the more-extended formalism supported by our current library) using the benchmark used in [23] to compare four outstanding research approaches in the CIG literature (Section 5). This is one of the most significant contributions of our paper, and is entirely new.
Section 6 contains some experiments we run to evaluate META-GLARE. Section 7 contains related works and comparisons, Section 8 describes our future works and Section 9 presents conclusions.
====
Also as suggest by Reviewer #2, we add the following sentence (before statements in lines 618-622, old version) in section 4 to stress the difference to previous work:
Notably, algorithms Algorithms 2-4 are an original contribution of this paper, being a substantial refinement and improvement of the “high-level” skeleton algorithm sketched in [16] and reported in Section 2.3 of this paper.
Additionally, in the revised version, we have also summarized the main goals and contributions of the paper also at the beginning of the Concluding Section, as reported below:
The overall goal of our work is the definition of META-GLARE, the first shell in the literature to facilitate the (formalism-based) design and development of CIG systems. While the general architecture of META-GLARE has been already published in [16], in this paper we focus on its execution engine. From the “technical” point of view, the main original contributions of our paper are:
- The algorithms constituting the basis of META-GLARE execution engine
- The library of control attribute types
- An evaluation of the expressiveness of our current formalism (META-GLARELib) using the benchmark in [23]
Additionally, some experiments we run to apply META-GLARE to use-cases are also presented.
====
1331-1372: Move this part to a separate section ‘Future Work’, before the Conclusions.
====
We have revised the Conclusion section and also created a new ‘Future Work Section (Section 8) reported below:
- Future works
The implementation of META-GLARE execution engine (HG_EXECUTION module in Fig. 1) is almost completed. In particular, the current implementation is more focused on management control-flows and not on issues concerning the interaction with the users-physicians. After the next implementation works, we plan to start as soon as possible an extensive experimental evaluation of our approach.
Notably, the current approach shows some limitations as discussed in Section 5.2, in our future work we also aim to further extend it, to support also new control structures and to overcome these limitations.
We plan to exploit META-GLARE the fast prototyping and the possibility of sharing attributes in our research, to develop CIG systems addressing new CIG tasks. In particular, we plan to exploit the META-GLARE approach in the field of CIG education; (see Experiment 3 in Section 6). We plan to build GLARE-Edu exploiting META-GLARE (see considerations in Section 6). This research will be done in AI-LEAP project, a project founded by Fondazione Compagnia di San Paolo and by Fondazione CDP, Bando Intelligenza Artificiale 2. Also, META-GLARE will allow us to support “personalized” CIGs in our projects with physicians. This means not only making the CIG systems (built using META-GLARE) more appealing, but META-GLARE will allow us to respond more easily and quickly to specific requests and issues domain-dependent and task-dependent.
===
1346: Explain what you mean by ‘almost’
===
In ‘Future Work’, we describe that the current state of implementation that is focused principally on the control flows management and not on user interaction issues. These aspects will be faced as the next step, since they are essential for the extensive experimentation, which we aim to do involving physicians and medical staff.
Round 2
Reviewer 2 Report
none
Reviewer 3 Report
The authors have satisfactorily addressed my concerns.
Minor editing of English language required.